# When to Intervene: Learning Optimal Intervention Policies for Critical Events

**Niranjan Damera Venkata** [*]
Digital and Transformation Organization
HP Inc., Chennai, India
niranjan.damera.venkata@hp.com

**Chiranjib Bhattacharyya**
Dept. of CSA and RBCCPS
Indian Institute of Science, Bangalore, India
chiru@iisc.ac.in

## Abstract

Providing a timely intervention before the onset of a critical event, such as a system failure, is of importance in many industrial settings. Before the onset of the critical event, systems typically exhibit behavioral changes which often manifest as stochastic co-variate observations which may be leveraged to trigger intervention. In this paper, for the first time, we formulate the problem of finding an optimally timed intervention (OTI) policy as minimizing the expected residual time to event, subject to a constraint on the probability of missing the event. Existing machine learning approaches to intervention on critical events focus on predicting event occurrence within a pre-defined window (a classification problem) or predicting time-to-event (a regression problem). Interventions are then triggered by setting model thresholds. These are heuristic-driven, lacking guarantees regarding optimality. To model the evolution of system behavior, we introduce the concept of a hazard rate process. We show that the OTI problem is equivalent to an optimal stopping problem on the associated hazard rate process. This key link has not been explored in literature. Under Markovian assumptions on the hazard rate process, we show that an OTI policy at any time can be analytically determined from the conditional hazard rate function at that time. Further, we show that our theory includes, as a special case, the important class of neural hazard rate processes generated by recurrent neural networks (RNNs). To model such processes, we propose a dynamic deep recurrent survival analysis (DDRSA) architecture, introducing an RNN encoder into the static DRSA setting. Finally, we demonstrate RNN-based OTI policies with experiments and show that they outperform popular intervention methods.

## 1 Introduction

We consider a problem that involves the monitoring of observations from a discrete-time stochastic process with the goal of intervening to prevent the costly occurrence of a critical event in the future. A critical event is missed when intervention is not triggered before the critical event occurs. Intervening too soon, before the critical event occurs, also incurs cost. In general, delaying an intervention in order to be more confident of the intervention decision leads to an increased probability of missing the event. The general goal is the design of algorithms that can be used to trigger a *timely* intervention. The objective is to intervene as late as possible, by minimizing residual time-to-event subject to a constraint on the probability of missing the event. We term this problem the Optimally Timed Intervention problem (OTI Problem). Note that we are concerned only with the timing of an intervention, not with its effectiveness in mitigating the occurrence of the critical event.

---

[*]Niranjan Damera Venkata was with the Artificial Intelligence and Emerging Compute Lab (AIECL) at HP Labs during part of this work.

36th Conference on Neural Information Processing Systems (NeurIPS 2022).

The OTI problem has significant practical significance in a number of fields. Critical events could be, for example, the failure of a machine or key component, customer churn, or a fall that could result in death. Contemplated interventions in these cases could be a machine or part replacement, sending a marketing offer, or a move to managed care, respectively. Intervening on such events ahead of time, in an optimal manner is of great significance to businesses and society, due to the potential of mitigating the impact of the occurrences of these events. However, the literature formally studying this problem has remained sparse.

## 1.1 Outline of contributions

The OTI problem is hard since evolution towards a critical event is complex with typically scarce occurrences of critical events. We make the following contributions to the study of the OTI problem:

- A process under observation may exhibit degradation/disorder as it evolves towards a critical event. We capture this feature by introducing the concept of a hazard rate process, which is a stochastic process modeling the evolution of conditional hazard rates as new observations are taken. A key contribution of our work is to formalize the OTI problem in equation (OTI problem) and show that the OTI problem is equivalent to an optimal stopping problem on the underlying hazard rate process (Proposition 3.1). This allows us to tap into optimal stopping theory [37, 9, 4] to unlock the structure of OTI policies and also leverage computational methods of estimating/learning hazard rate functions. These are key to our development of a general machine learning approach to the OTI problem.

- Under Markovian assumptions on the hazard rate process, we show that the optimal intervention policy decision rule at each time step can be analytically determined by the conditional hazard rate function at that time step (Theorem 4.1). Further we show that the optimal policy reduces to a variable threshold on the expected time-to-event(Corollary 4.1.1), showing that static threshold policies are sub-optimal in general.

- We show that our theory applies to an important class of hazard rate processes generated by recurrent neural networks (Section 5). We term these processes *neural* hazard rate processes. We may therefore apply survival analysis methods to learn such neural hazard rate processes and analytically compute optimal intervention policies for critical events. We adapt the state of the art deep dynamic survival analysis (DRSA) work [29] to our setting. Unlike DRSA, our dynamic DRSA (DDRSA) setting requires modeling dynamic hazard rate functions. We do this by adapting the DRSA architecture to our dynamic covariate evolution setting using an encoder-decoder sequence-to-sequence (seq2seq) architecture.

## 2 Optimally Timed Interventions (OTI) on critical events

### 2.1 Notation

We represent the monitored stochastic process by a discrete, temporal sequence of random variables with the random vector $\mathbf{X}_j$ representing a d-dimensional real-valued observation at time step $j$. Let the $d \times j$ random matrix $\mathcal{X}_j = [\mathbf{X_1}, \mathbf{X_2}, \cdots \mathbf{X}_j]$ represent the observation history until time step $j$. The critical event is assumed to occur at a random time-step $L \leq L_{max}$[2], $L_{max} \in \mathbb{N}$. An intervention at time step $j$ is considered early if the intervention occurs at a time-step $j < L$. We consider interventions at $j \geq L$ late, by definition. We assume, without loss of generality, that interventions are instantaneous. If a fixed lead time of $\delta$ time-steps is required for an intervention, we may simply consider the random variable $L' = L - \delta$ as the event occurrence time-step and seek timely intervention before $L'$. For any process realization, at a time-step $j$, we assume that we know with certainty, if the critical event has already occurred at a prior time-step or not. We define $\pi_j := \mathbb{1}(L < j | \mathcal{X}_j) = \mathbb{1}(L < j | \mathbf{X}_j)$. $\pi_j$ serves to indicate if the critical event has already occurred ($\pi_j = 1$) or not ($\pi_j = 0$) prior to time step $j$, given $\mathbf{X}_j$.

Given a dataset of process trajectories along with corresponding critical event occurrences, we seek intervention strategies for *timely* intervention. We consider data-driven sequential decision rules (intervention policies) that are defined as follows:

---

[2]This upper bound assumption does not pose practical limitations due to our survival analysis approach. We may treat unobserved critical events as censored. See Section 5.1 for details

**Definition 2.1** (Intervention policies). An intervention policy is a sequence of functions $\varphi = (\varphi_1, \varphi_2, \cdots \varphi_j, \cdots)$, where $\varphi_j : \mathbb{R}^{d \times j} \mapsto \{0, 1\}$ maps the history of observations at a time step $j$ to a decision of whether to intervene ($\varphi_j(\mathcal{X}_j) = 1$) or not ($\varphi_j(\mathcal{X}_j) = 0$).

We define $\varphi_{L_{max}-1}(\mathcal{X}_{L_{max}-1}) := 1$ to ensure that we have at least one decision to intervene before $L_{max}$. Thus, for technical completeness, we assume observations can be taken until $L_{max} - 1$. In cases where the occurrence of the critical event makes it impossible to collect further observations in practice, we can can simply fill such observations with dummy values (ex: zeros) without any loss of generality since these observations are not predictive of the critical event, and therefore do not feature in computing optimal intervention decisions (see discussion of Proposition 3.1).

The intervention policies we consider in this paper are *terminal*, so only a single intervention is contemplated on each process realization. Once the first decision to intervene is triggered on a given realization of the monitored process, the policy *effectively* stops, i.e. an intervention action is assumed to be initiated and all future policy triggers for that process realization are ignored. Note however, that we can allow for policy reactivation after recovering from the critical event. The observations starting at such policy reactivations are treated as independent realizations of the process. So if we recover from the occurrence of a first critical event and re-activate the policy to to monitor for a second second critical event, these cases will be considered as independent trials.

**Definition 2.2** (policy stopping time). Intervention on a monitored process happens at a random time step $\tau := \min\{0 < j < L_{max} : \varphi_j(\mathcal{X}_j) = 1\}$. We call this time the stopping time [3] of the policy.

## 2.2 Optimally timed intervention (OTI) on critical events

We now formally state the optimally timed intervention (OTI) problem for the first time. We seek to find intervention policy (or equivalently, the stopping time) that minimizes expected residual time-to-event, defined as $\mathbb{E}[(L - \tau)^+]$ subject to the constraint that the probability of missing the critical event is at most $\alpha \in (0, 1]$:

$$\tau^* = \underset{\tau \in \mathcal{M}_\alpha}{\arg\inf} \, \mathbb{E}[(L - \tau)^+] \text{ where: } \mathcal{M}_\alpha = \{0 < \tau < L_{max} : \mathbb{P}(L \leq \tau) \leq \alpha\} \tag{1}$$

The expected residual time-to-event is a variant of mean residual life (MRL), a quantity that is well studied in literature with application in multiple domains [1, 14]. Note that there exists a multiplier $C_\alpha$, such that the OTI problem may be expressed in the following equivalent, unconstrained form:

$$\tau^* = \underset{0 < \tau < L_{max}}{\arg\inf} \, \mathbb{E}[(L - \tau)^+] + C_\alpha \mathbb{P}(L \leq \tau) \text{ for some } C_\alpha > 0 \tag{OTI problem}$$

The objective encodes the natural trade-off between expected residual time-to-event, and the probability of missing the critical event: as one attempts to intervene closer to the event, the probability of missing the event goes up. $C_\alpha$ can directly be interpreted as the risk (cost) of missing the critical event. Changing $C_\alpha$ allows one to vary the risk trade off between the miss-rate and expected time-to-event. Note that the theory allows observations after an intervention in the problem formulation. So an intervention policy can consider these covariates and fire multiple times, however, the OTI objective function in equation (OTI problem) is designed to consider only the first policy trigger due to the fact that the stopping time $\tau$ is set based on the first policy trigger (see definition 2.2).

A related objective also appears in optimal stopping formulations for Bayesian quickest change point detection problem [35], famously called the Shirayaev problem [27], where the objective is to monitor a stochastic process for the quickest possible detection of a change point or stochastic disorder. Here the trade-off is between stopping early (a false alarm), before a change point manifests, versus incurring the expected delay in detecting a change point, after it has occurred. The goal is to minimize expected detection time delay from the true change point subject to a constraint on the probability of a false alarm.

---

[3]a stopping time in probability theory is random a variable $\tau$ such that $\mathbb{1}(\tau = j)$ is a function of $\mathcal{X}_j$. So we can determine if $\tau = j$ or not by only considering $\mathcal{X}_j$ [27].

## 2.3 Related work

Existing widely used approaches to timely intervention on critical events focus on predicting event occurrence with either a classification or regression problem formulation. Interventions are triggered by tuning model thresholds, a process which lacks any guarantee on optimality:

**Window based critical event prediction:** Intervention is a binary decision. At each time-step a decision must be made to intervene or not. In window based intervention (WBI) this is often modeled directly as a binary classification problem where we look for event occurrence within a specified future window. A classifier may then be learnt to predict the probability of event occurrence in the window. An optimal threshold on this probability is then tuned to choose an optimal intervention trigger threshold. WBI policies are standard practice in predictive maintenance[31, 2, 20, 2, 36, 38, 13].

**Time-to-Event prediction (Survival analysis):** Classical survival analysis deals with predicting the time to an event (ex: death, machine failure) given a vector of observations (co-variates) obtained from a subject in the presence of incomplete (censored) observations [21]. The goal is to estimate functions such as the hazard rate (rate of death as a function of time) or survival function (probability of survival past a given time) and understand the impact of co-variates on these functions. Classical parametric models [10, 39] have been extended in recent work using machine learning approaches [40, 17, 5, 28, 19] including approaches that extend survival analysis to model the time-varying effects of covariates [33, 3]. These have deep learning counterparts [29, 12] that use recurrent neural networks to model the survival function. Recent work has applied modern deep learning approaches to longitudinal data which are sequential observations taken over time for each subject. [15, 18, 16]. Corresponding work also exists in the area of machine failure prediction [24]. These methods are complementary to the primary goal of this paper where we focus on obtaining optimal intervention policies. In practice, intervention policies based on survival analysis are typically just thresholds on the predicted time-to-event.

## 3 OTI as an optimal stopping problem

**Definition 3.1** (Optimal Stopping Problem)**.** A problem of the form: $\tau^* = \arg\inf_\tau \mathbb{E}\left[R_\tau(X_\tau)\right]$ where $R_j(\cdot)$ is a risk function, $\tau$ is a stopping time, and $\{X_j\}$ is a stochastic process.

Optimal stopping problems [37, 9, 4] are a subclass of stochastic control problems with only two possible actions, stop and continue. The goal is to determine when to stop, based on the observations, so that overall expected cost/risk over possible stopping times, is minimized. Optimal stopping techniques have been applied to several problems such as sequential hypothesis testing [37, 11, 7] (the sequential probability ratio test (SPRT)), early change point detection [27, 35, 34, 30] (the Shirayaev problem), early classification [23] and reliability theory [6, 22]. However, the approaches in such cases are limited to simple probabilistic models of machine degradation with a single or few co-variates. General model-free reinforcement learning approaches to the optimal stopping problem are generally feasible in situations where we have a lot of data or in a model-based setting where the underlying randomness can efficiently be simulated and the model can be used to draw Monte-Carlo samples [8]. This is not an option in most critical event prediction settings when the critical event data is often scarce and no simple model exists for the evolution of the co-variates.

In this section we formulate the OTI problem as an optimal stopping problem. Our formulation of the OTI problem in this section, establishes, for the first time, a link between the OTI problem, survival analysis (allowing the modeling of complex co-variate evolution with impact on critical-event occurrence) and optimal stopping theory. We model the dependence of the critical event time on the monitored covariate observations by hazard rate functions conditioned on observation history. To this end, we will use the following definitions of a conditional hazard rate function, $h_j(\cdot)$, and conditional survival function, $S_j(\cdot)$:

$$h_j(k) := \mathbb{P}(L = j + k | \mathcal{X}_j, L \geq j) \qquad S_j(k) := \mathbb{P}(L > j + k | \mathcal{X}_j, L \geq j) \qquad (2)$$

Note that we have omitted the implied conditioning on the covariates in our notation for $h_j(k)$, purely for convenience. We now define the concept of a hazard rate process that encodes the relationship between the evolution of co-variates and onset of the critical event.

**Definition 3.2** (Hazard Rate Process)**.** Consider the random vector constructed from the conditional hazard rate function $h_j(\cdot)$ given by $\mathbf{H}_j = [h_j(0), h_j(1) \cdots h_j(L_{max} - 1)]^T$. We call the sequence of tuples $\{(\mathcal{X}_j, \mathbf{H}_j)\}$ a hazard rate process.

### 3.1 OTI as an optimal stopping problem on a hazard rate process

**Proposition 3.1** (OTI on a hazard rate process). Given a hazard rate process $\{(\mathcal{X}_j, \mathbf{H}_j)\}$, the optimal critical event intervention problem of equation (OTI problem) reduces to the following optimal stopping problem on the corresponding hazard rate processes:

$$\tau^* = \operatorname*{arg\,inf}_{0<\tau<L_{max}} \mathbb{E}\left[(1-\pi_\tau)R^H(\mathbf{H}_\tau) + \pi_\tau C_\alpha\right] := \operatorname*{arg\,inf}_{0<\tau<L_{max}} \mathbb{E}\left[R(\pi_\tau, \mathbf{H}_\tau)\right] \tag{3}$$

where $R^H(\mathbf{H}_\tau)$ is the intervention risk of stopping at time $\tau$, given that the critical event has not already occurred prior to $\tau$, given by:

$$R^H(\mathbf{H}_\tau) = \sum_{k=0}^{L_{max}-1} \prod_{m=0}^{k} (1 - h_\tau(m)) + C_\alpha h_\tau(0) \tag{4}$$

*Proof.* We only give a proof sketch here. The proof of the theorem is given in Appendix A. Starting with terms in the OTI problem: for residual time to event we may write $(L-\tau)^+ = \sum_{k\geq 0} \mathbb{1}(L > \tau+k)$, and then use the law of iterated expectations, $\mathbb{E}[(L-\tau)^+] = \mathbb{E}[\sum_{k\geq 0} \mathbb{E}[\mathbb{1}(L > \tau+k)|\mathcal{X}_\tau, L \geq \tau](1-\pi_\tau)]$. We see that this expression is $\mathbb{E}[(1-\pi_\tau)\sum_{k\geq 0} S_\tau(k)] = \mathbb{E}[(1-\pi_\tau)\sum_{k\geq 0} \prod_{m=0}^{k}(1-h_\tau(m))]$. Similarly, we have $\mathbb{P}(L \leq \tau) = \mathbb{E}[(1-\pi_\tau)h_\tau(0) + \pi_\tau]$. Therefore the OTI objective may be expressed as $\mathbb{E}[R(\pi_\tau, \mathbf{H}_\tau)] = \mathbb{E}[(1-\pi_\tau)R^H(\mathbf{H}_\tau) + \pi_\tau C_\alpha]$. $\qquad\square$

Note that the formulation of equations (3) and (4) exposes the dependence of the OTI objective on the co-variates, since $h_\tau(.)$ depends on the covariates. This was implicit in the original formulation in the previous section. A key observation is that if the critical event has already occurred (when $\pi_\tau = 1$), the objective reduces to a constant ($C_\alpha$), independent of the co-variates, and policy actions have no influence on the objective. Hence, we may effectively restrict our attention to the case $\pi_\tau = 0$.

## 4 Analytic solution to the OTI problem

In order to solve the OTI optimal stopping problem we make the assumption that the hazard rate processes we deal with satisfy the Markov property:

**Definition 4.1** (Markov hazard rate process). Consider the hazard rate process $\{(\mathcal{X}_j, \mathbf{H}_j)\}$. We call it a Markov hazard rate process, if we have:

$$\mathbb{P}(\mathbf{X}_{j+1}|\mathcal{X}_j) = \mathbb{P}(\mathbf{X}_{j+1}|\mathbf{X}_j) \;\; \forall j \tag{5}$$

As a consequence, we can write $\mathbf{H}_j = f_j(\mathbf{X}_j)$ where $f_j(\cdot)$ is a function encoding the dependence of the hazard rates on the co-variates at time-step $j$. With this assumption in place, we have:

**Theorem 4.1** (OTI policy for a Markov hazard rate process). *Given a Markov hazard rate process* $\{(\mathcal{X}_j, \mathbf{H}_j)\}$ *and associated risk process* $\{R^H(\mathbf{H}_j)\}$, *the optimal intervention policy that solves the problem posed in Proposition 3.1 is a policy* $\varphi^*$ *that calls for intervention at stopping time* $\tau^* = \min\{0 < j < L_{max} : \varphi_j^*(\mathcal{X}_j) = 1\}$ *where:*

$$\varphi_j^*(\mathcal{X}_j) = \begin{cases} \mathbb{1}(R^H(\mathbf{H}_j) \leq V_j(\mathbf{H}_j)), & \text{if } j < L_{max} - 1 \text{ and } \pi_j = 0 \\ 1, & \text{else} \end{cases} \tag{6}$$

*with continuation value functions* $V_j(\mathbf{H}_j)$ *given by:*

$$\min_{0<l<L_{max}-j} \sum_{k=0}^{L_{max}-l-1} \prod_{m=0}^{k+l} (1 - h_j(m)) + C_\alpha h_j(0) + C_\alpha \sum_{m=1}^{l} \left[\prod_{k=0}^{m-1} (1 - h_j(k))\right] h_j(m) \tag{7}$$

*Proof.* We only give a proof sketch here. The complete proof of the theorem is given in Appendix A. The key is to notice that under the Markov hazard rate process assumption, and the form of of equations (3) and (4), that risk $\tilde{R}_j = R(\pi_j, \mathbf{H}_j)$ is a function of $\mathbf{X}_j$, so we have an optimal stopping problem that has a so-called Markov Representation [27]. The optimal policy has can be expressed as $\varphi_j^*(\mathcal{X}_j) = \mathbb{1}(\tilde{R}_j \leq \tilde{V}_j(\mathcal{X}_j))$ with $\tilde{V}_j(\mathcal{X}_j) = \min_{l>0} \mathbb{E}[\tilde{R}_{j+l}|\mathbf{X}_j]$. So we need only compute

$\mathbb{E}[\tilde{R}_{j+l}|\mathbf{X}_j]$, which is the expected future risk over the possible occurrences of the critical event for a policy that stops at $j + l$. When $\pi_j = 0$, we have $\tilde{R}_j = R^H(\mathbf{H}_j)$. Further, reasoning using the hazard rates $h_j(k)$, we calculate $\mathbb{E}[\tilde{R}_{j+l}|\mathbf{X}_j]$ analytically giving rise to the final form for $\tilde{V}_j(\mathcal{X}_j)$ which we express as $V_j(\mathbf{H}_j)$. When $\pi_j = 1$, $\tilde{R}_j = \min_{l>0}\mathbb{E}[\tilde{R}_{j+l}|\mathbf{X}_j] = C_\alpha$, so it is optimal to trigger intervention, setting $\varphi_j^*(\mathcal{X}_j) = 1$. Finally, we have $\varphi_{L_{max}-1}^*(\mathcal{X}_j) = 1$, by definition, giving rise to the final form of the optimal policy. $\qquad\square$

Thus, the optimal policy may be constructed explicitly with knowledge of only the conditional hazard rate functions at that time. The *continuation value* function $V_j(\cdot)$ gives the minimum expected cost of continuing to take observations, having obtained a conditional hazard rate function ($\mathbf{H}_j$) at time $j$. It is clear that the the optimal policy is a time-varying threshold on the risk process in general.

The following Corollary allows a further important simplification of the optimal policy:

**Corollary 4.1.1** (OTI policy as a time varying threshold on time-to-event). *Given $\pi_j = 0$, let $T_j$ be the expected residual time-to-event at time $j$. The optimal OTI policy of Theorem 4.1 may be reduced to the following form:*

$$\varphi_j^*(\mathcal{X}_j) = \begin{cases} \mathbb{1}(T_j \leq V_j'(\mathbf{H}_j)), & \textit{if } j < L_{max} - 1 \textit{ and } \pi_j = 0 \\ 1, & \textit{else} \end{cases} \tag{8}$$

*where,* $V_j'(\mathbf{H}_j) = \min_{0<l<L_{max}-j} \sum_{k=0}^{L_{max}-l-1} \prod_{m=0}^{k+l}(1 - h_j(m)) + C_\alpha \sum_{m=1}^{l}\left[\prod_{k=0}^{m-1}(1 - h_j(k))\right]h_j(m)$

$$\tag{9}$$

*with,* $T_j = \sum_{k=0}^{L_{max}-1} \prod_{m=0}^{k}(1 - h_j(m)) = \sum_{k=0}^{L_{max}-1} S_j(k) \tag{10}$

*Proof.* The full proof is given in Appendix A. We note that the immediate stopping risk for time $j$ can be written as: $\tilde{R}_j = T_j + C_\alpha h_j(0)$. Further, the second term, $C_\alpha h_j(0)$, may be canceled from both sides of the inequality defining the optimal stopping policy, resulting in the simplified rule. $\quad\square$

This Corollary clearly shows that static threshold TTE policies are sub-optimal in general, since $V_j'(\mathbf{H}_j)$ may vary with time index $j$. $V_j'(\mathbf{H}_j)$ contains two terms. The first term, which is the risk of stopping early, before the critical event, intuitively gets smaller as $j$ progresses toward the critical event. The second term is the risk of missing the critical event that depends on the cost of missing the event. It is high where the onset of the critical event is anticipated.

## 5 Neural hazard rate processes

The Markov assumption is not a significant restriction since any co-variate process may be made Markovian, by including all relevant information from the past into the current state (at the cost of increasing the dimension of the problem). While one may adapt any survival analysis technique (in theory) that estimates conditional hazard rate functions and apply our theory, recurrent neural networks (RNNs) are a good fit for being able to model complex patterns in multi-variate time-series data and for summarizing observation history.

The class of hazard rate processes $\{(\mathcal{Z}_j, \mathbf{H}_j)\}$ that can be modeled by recurrent neural networks with hidden-states $\mathbf{Z}_j$ and conditional hazard rates given by $\mathbf{H}_j = f(\mathbf{Z}_j)$ for some function $f(\cdot)$ of the hidden states), satisfy our Markov hazard rate process assumptions. Indeed, from the RNN dynamics, the probability of the next hidden state is conditionally independent of the past hidden states, given the current hidden state. Specifically, we have: $\mathbb{P}(\mathbf{Z}_{j+1}|\mathcal{Z}_j) = \mathbb{P}(\mathbf{Z}_{j+1}|\mathbf{Z}_j), \forall j$. Note, however, that we do not require the co-variate evolution to be Markovian. The hazard rate functions can depend on the past history of co-variates in general. RNNs compute feature transformations of past covariate history (prior to $j$) and seek to embed all required information about the past into $\mathbf{Z}_j$, so a hazard rate function that is computed from $\mathbf{Z}_j$, can be quite complex and include (to the extent it is present in $\mathbf{Z}_j$) information from past covariates.

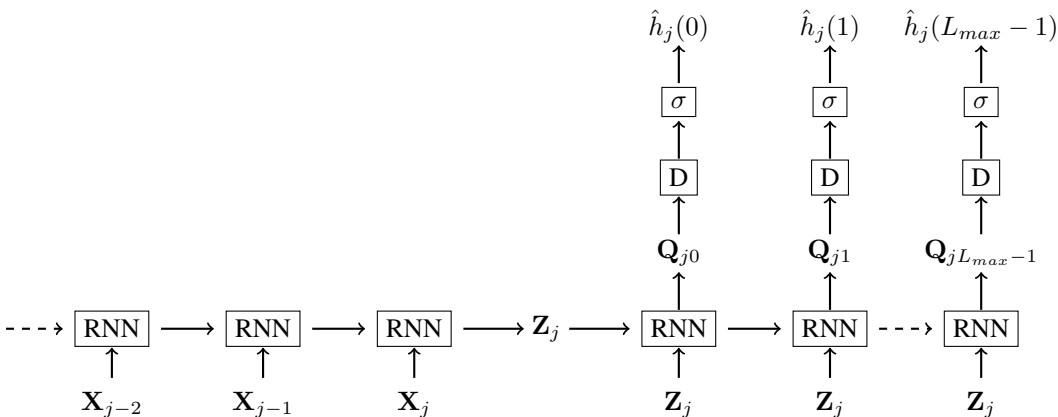

Figure 1: Architecture diagram for Dynamic Deep Recurrent Survival Analysis (DDRSA).

### 5.1 Dynamic Deep Recurrent Survival Analysis (DDRSA)

Since the OTI theory developed in the previous section directly applies to neural neural hazard rate processes, we may fit a neural hazard rate process model to data and then apply the optimal OTI policy to intervene. Thus, our approach requires estimates of conditional hazard rates, that evolve over time (hazard rate processes).

Ren et. al. [29] developed Deep Recurrent Survival Analysis (DRSA), a deep recurrent neural architecture that uses and RNN to map a static covariate vector to a single conditonal hazard rate function (so it is not a hazard rate process that evolves over time). Unlike DRSA, our Dynamic DRSA (DDRSA) setting requires modeling dynamic survival and hazard rate functions. We do this by adapting the DRSA architecture to our dynamic covariate evolution setting. While there are other attempts at modeling the dynamic evolution of survival functions [18, 24] they either make distributional assumptions [24] or have difficulty with long sequences due to large softmax layer [18].

Figure 1 shows our DDRSA architecture which is a classic encoder-decoder architecture. DDRSA uses an encoder RNN that maps co-variate history $\mathcal{X}_j$ to an encoder hidden state vector $\mathbf{Z}_j$. This vector is then passed into a decoder DRSA-RNN architecture, producing decoder hidden states $\mathbf{Q}_{jk}$ and corresponding conditional hazard rates $h_j(k) = \sigma(D(\mathbf{Q}_{jk}))$, where $\sigma(\cdot)$ represents the sigmoid activation function and $D(\cdot)$ represents a Dense layer. DDRSA can also be seen as a generative model for a *neural* hazard rate process.

During training, each training sample, randomly selected from one of the time-series at time step $j$ includes a look-back window of co-variates $\mathbf{X}_{j-K} \cdots \mathbf{X}_j$ that are fed into the encoder input. The encoder's final hidden state is replicated and provided to the decoder as input to each time step. We leverage the DRSA loss of Ren et. al. [29] which is composed of the following loss components for each sample:

$$l_z = \prod_{k=0}^{l-1} \left[1 - \hat{h}_j(k)\right] \hat{h}_j(l) \, , \, l_u = \left[1 - \prod_{k=0}^{L_{max}-1} (1 - \hat{h}_j(k))\right] \, , \, l_c = \prod_{k=0}^{L_{max}-1} (1 - \hat{h}_j(k)) \quad (11)$$

$l_z$ is the likelihood of the critical event occurrence at at time $j + l$, assuming that the event is not censored. $l_u$ is the likelihood of event occurrence (a.k.a. event rate) at any future time in the decoder output, given that the sample is not censored. $l_c$ gives the likelihood of a censored sample, when the event does not occur at a future time over the decoder time-steps. The final loss for a sample is :

$$L_f = -\lambda \log l_z - (1 - \lambda)\left[(1 - c) \log l_u + c \log l_c\right] \quad (12)$$

where $\lambda$ is a trade off fraction and $c$ is the censoring indicator, with $c = 1$ for a censored sample and $c = 0$ for uncensored. Losses averaged over a mini-batch of samples are used to learn the survival model. The sequential DRSA loss allows effective supervision at each time-step allowing it to be applied to model long survival functions without resorting to restrictive parametric distributions.

During inference, we estimate, at any time-index $j$, $h_j(0), h_j(1), \cdots h_j(L_{max} - 1)$ as decoder outputs by passing values of co-variates $\mathbf{X}_{j-K} \cdots \mathbf{X}_j$ into the DDRSA architecture encoder. Then,

we estimate the expected residual time-to-event using equation (10) and the threshold function using equation (9). Intervention is triggered at time $j$ if the estimated expected residual time-to-event falls below the threshold function.

Note that the model architecture is such that we can generate a long sequence of hazard rates. During training time, we restrict the decoder time-steps to a fixed value ($M \leq L_{max}$) and consider events appearing outside this implied range as censored. However, during inference, the RNN is unrolled over the full $L_{max}$ time-steps. To allow for such unrolling, we eschew explicit decoder time-step inputs, instead forcing the decoder to learn to increment time-index. Providing the decoder time-step indices explicitly as inputs to the decoder as is done in the original DRSA-RNN [29] architecture, is not desirable, because, when we unroll the RNN to $L_{max}$ time-steps, we would need to give the decoder inputs it has not seen during training. Such flexible handling of long sequences is a key practical advantage of the DDRSA architecture and our survival formulation.

## 6 Experiments

The goal of this section is twofold:

1. To complement the theory developed in the earlier sections with examples to validate and build intuition into survival and hazard rate processes how the optimal policy works to produce interventions. (Section 6.1)

2. To demonstrate that the theory does indeed translate into an effective practical methodology embodied by the DDRSA-RNN that can model hazard rate processes and generate interventions using the OTI policy. We call this combined approach the OTI-DDRSA-RNN. We show that OTI-DDRSA-RNN outperforms static threshold policies that are either window based (WBI-RNN) or time-to-event based (TTE-RNN)(Section 6.2).

Experiments are performed on two real-world datasets to validate our goals. Experiments on a third synthetic dataset are included in Appendix C.

**Turbofan Engine Failure Data [26, 32]:** This dataset from NASA provides train and test data to predict Turbofan engine failures. The training set has simulated multivariate time-series observations from 21 sensors on a fleet of 218 engines. Each engine operates normally at the start and develops a fault at some point resulting in eventual failure. The test set consists of 218 partial engine time-series with remaining useful life (in engine cycles) indicated in each case. We could not use the test data as provided since it is missing part of every time series close to failure.

**Azure Predictive Maintenance Guide Data[25]:** This is a dataset from a guide provided by Microsoft as a case study in failure prediction. The data comes from multiple sources including time-series of voltage, rotation, pressure and vibration measurements collected from 100 machines in real time averaged over every hour, error logs, machine information (type, age etc.) and 720 failure records. We adopt the same pre-processing, feature generation as given in the guide. Also, we down-sample the data to take one sample every 15 hours.

For both datasets, we train on 70% of randomly selected co-variate time-series sequences and hold out 30% of the sequences for testing. From the training set a further 30% of the sequences are set aside as validation data to tune model parameters and policy thresholds. This process is repeated to produce 10 random train-validation-test splits. Observations are normalized with min-max transformation that transforms training inputs to lie in the range $[-1, 1]$. Model selection details and parameter settings are discussed in Appendix B.

### 6.1 Discussion of OTI interventions on real data

We fit the DDRSA-RNN on the covariate time-series leading upto failure and predict on a single test set sequence. This produces, at each time step $j$ of the sequence, an estimated conditional hazard rate vector $\mathbf{H}_j$ (representing the conditional hazard rates into the future), which gives us a probabilistic estimate of the onset of the critical event at each time step, taking into account co-variate history upto that time step. Figure 2(a) shows the evolution of estimated conditional hazard rate functions. We see that the onset of the critical event is clearly reflected in the progression of the estimated vectors $\mathbf{H}_j$.

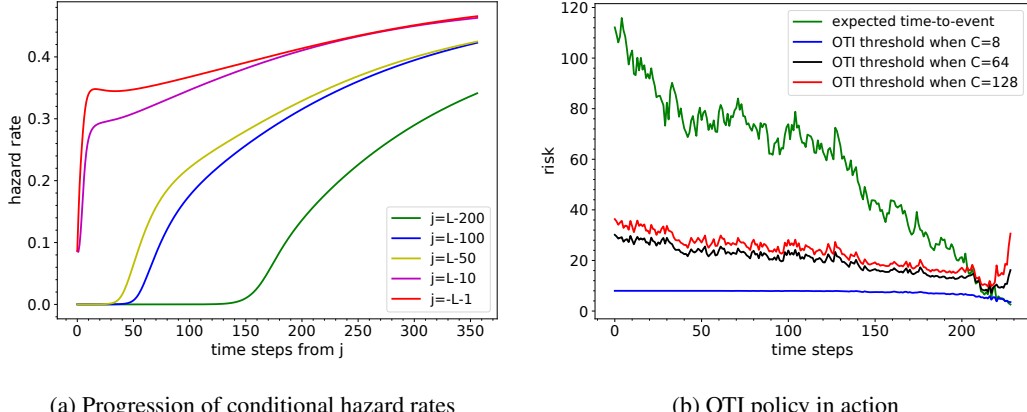

(a) Progression of conditional hazard rates        (b) OTI policy in action

Figure 2: (a) Example of typical evolution of estimated conditional hazard rates estimated by DDRSA as $j$ approaches the critical event. (b)The OTI policy in action. The OTI policy calls for intervention when the estimated time-to-event falls below a time-varying policy threshold. As the cost $C$ of missing the critical event increases, interventions are triggered earlier.

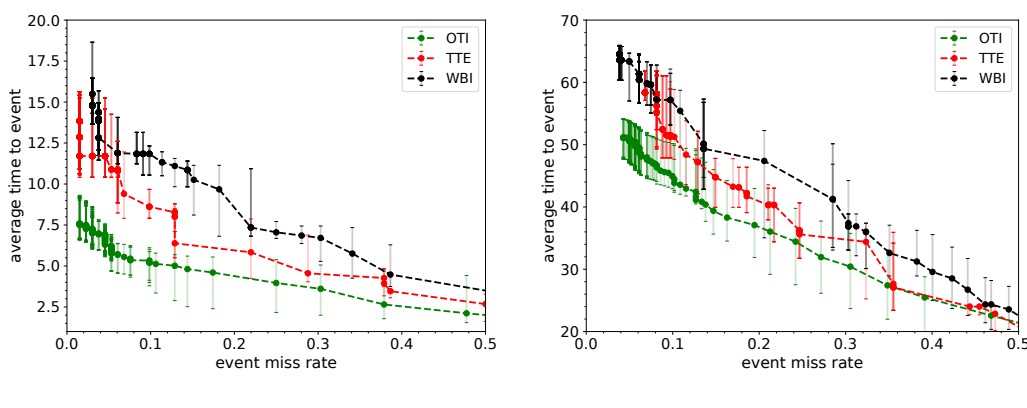

(a) NASA turbofan data: policy trade-off plot    (b) Azure PM guide data: policy trade-off plot

Figure 3: Performance comparison of OTI-DDRSA-RNN, WBI-RNN and TTE-RNN policies. Trade-off plots showing the median (solid circle) and inter-quartile range (vertical bars) for each $C_\alpha \in \{8, 10, \cdots, 256\}$ computed over 10 randomized test folds.

Next, we demonstrate the working of the OTI-policy. Figure 2(b) plots the estimated time-to-event at each time step, estimated from $\mathbf{H}_j$ using equation (10). We also plot the adaptive threshold $V_j^{'}(\mathbf{H}_j)$, calculated using equation (9) of Corollary 4.1.1. An intervention is triggered when our time-to-event estimate falls on or below the adaptive threshold. Consider what happens as the cost of missing the critical event is varied. First, note that since time-to-event is independent of $C_\alpha$, only the threshold changes. Increasing the cost of missing the critical event increases the threshold causing earlier intervention as we would expect.

## 6.2 Performance Evaluation

In this section we will evaluate the performance of the proposed OTI-DDRSA-RNN vs. two static threshold policy baselines: A window based intervention RNN (WBI-RNN) and a time-to-event based intervention RNN (TTE-RNN). All RNNs share the same encoder architecture which is an LSTM with 128 step look-back and hidden state dimension of 16 units.

**WBI-RNN architecture**: The WBI-RNN has the same encoder as the DDRSA-RNN with the final encoder output $\mathbf{Z}_j$ in Figure 1 passed to a dense (fully connected) layer to produce a single output probability per training sample. We train the model with a binary cross-entropy loss between the

probability output and the binary state of event in a future window $W$ of time steps. A threshold is used on this probability to trigger interventions once the probability is greater than the threshold. The window size and threshold are selected by cross-validation (see Appendix B).

**TTE-RNN architecture**: The TTE-RNN uses the DDRSA architecture shown in Figure 1 and is learnt with the DRSA loss as discussed above. However, interventions are triggered with a static threshold that is set via cross-validation similar to the WBI-RNN.

All methods were trained, validated and tested using 10 randomized train-validation-test splits. Model parameter initialization seeds are also randomized across trials. Results reported are medians over the 10 test folds. WBI and TTE thresholds are tuned (individually, for 125 different settings of $C_\alpha \in \{8, 10, \cdots 256\}$) on each validation set using an empirical intervention policy risk:

$$R_{emp}(\varphi) = \underbrace{\frac{1}{N} \sum_{i=1}^{N} (L_i - \tau_i) \mathbb{1}(L_i \geq \tau_i)}_{\text{average time-to-event}} + C_\alpha \underbrace{\frac{1}{N} \sum_{i=1}^{N} \mathbb{1}(L_i \leq \tau_i)}_{\text{event miss-rate}} \tag{13}$$

where $(L = L_i, \tau = \tau_i)$ are realized outcomes for the $i^{th}$ series.

**Computational setup and inference time:** All experiments were run on a Tensorbook Laptop with 32GB RAM and having a single NVIDIA GeForce GTX 1070 with Max-Q GPU. Training on a single dataset for 10 folds and 125 cost points took around 2 hours. DDRSA inference time per time-step time was around 12 milliseconds. The corresponding OTI policy evaluation time per time-step was around 0.5 milliseconds.

Figure 3 summarizes the evaluation results in the form of policy trade-off curves. Each point represents a plot of the median average time-to-event vs the median event miss-rate, calculated for a particular setting of $C_\alpha$ (the corresponding inter-quartile ranges are also shown as vertical bars). Regarding the trade-off plots, a lower frontier is preferred, since it implies a lower average time-to-event for a given miss-rate and a lower miss-rate for a given average time-to-event. We see that the OTI-DDRSA-RNN outperforms the static threshold policies on both datasets.

**Limitations:** The quality of the OTI policy depends on the accuracy of the underlying survival analysis. Note that our approach requires fitting a survival model for $h_j(k)$ at each time step over several values of $k$. It is possible that some of these are better estimated than others. Also, in contrast, a threshold algorithm can be tuned for a specific $C_\alpha$ on a validation set while there is no $C_\alpha$ specific tuning for the OTI policy. Also, there is no direct way of setting $C_\alpha$ to guarantee a particular $\alpha$. We are forced to try a range of values for $C_\alpha$ and pick the values with most acceptable trade-offs.

## 7 Conclusions

We consider the task of monitoring a system for the occurrence of a critical event, by observing co-variates over time. The goal is to intervene in a timely manner. We formalize this for the first time as the Optimally Timed Intervention (OTI problem) which seeks a policy that triggers interventions such that for a given probability of missing the event, the expected residual time-to-event is minimized. We show that the OTI problem may be posed as an optimal stopping problem on a hazard rate process, which may be solved analytically under Markovian assumptions. A significant new insight is that the optimal policy at any time is an analytically computable threshold on expected time-to event that depends only on the conditional hazard rate function at that time. Finally, based on the theory we develop a practical RNN architecture (the OTI-DDRSA-RNN) to produce optimal interventions. Future directions include an analysis of the impact of inaccuracies in estimating the hazard rate process on the resulting intervention policy and the ability to focus the estimation of the hazard rate process to achieve optimal performance for a specified event miss-rate.

## 8 Acknowledgments

This work was initiated under a research agreement between HP Inc. and the Indian Institute of Science (IISc). We acknowledge Jerry Liu, M. Anthony Lewis, Matt Ellis, Scott Hallworth and Jennifer Hill at HP Inc. for their support and sponsorship of this work. Finally, we thank all the reviewers for their insightful comments that have helped improve the paper.

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
