# A Appendix A: Proofs

## A.1 Proof of Proposition 3.1

We first prove the following Lemmas:

**Lemma A.0.1.** $\mathbb{E}[(L - \tau)^+] = \mathbb{E}[(1 - \pi_\tau) \sum_{k=0}^\infty S_\tau(k)] = \mathbb{E}[(1 - \pi_\tau) \sum_{k=0}^\infty \prod_{m=0}^k (1 - h_\tau(m))]$

*Proof.* Note that we can write

$$
\begin{align}
(L - \tau)^+ &= \sum_{k \geq 0} \mathbb{1}(L - \tau > k) \tag{14} \\
&= \sum_{k \geq 0} \mathbb{1}(L > \tau + k) \tag{15}
\end{align}
$$

Taking expectations and using the law of iterated expectations we have:

$$
\begin{align}
\mathbb{E}[(L - \tau)^+] &= \mathbb{E}\left[ \sum_{k \geq 0} \mathbb{E}[\mathbb{1}(L > \tau + k)|\mathcal{X}_\tau] \right] \tag{16} \\
&= \mathbb{E}\left[ \sum_{k \geq 0} \mathbb{P}(L > \tau + k|\mathcal{X}_\tau) \right] \tag{17} \\
&= \mathbb{E}\left[ \sum_{k \geq 0} (1 - \pi_\tau)\mathbb{P}(L > \tau + k|\mathcal{X}_\tau, L \geq \tau) + \pi_\tau \mathbb{P}(L > \tau + k|\mathcal{X}_\tau, L < \tau) \right] \\
&= \mathbb{E}\left[ \sum_{k \geq 0} (1 - \pi_\tau)\mathbb{P}(L > \tau + k|\mathcal{X}_\tau, L \geq \tau) \right] \tag{18} \\
&= \mathbb{E}\left[ (1 - \pi_\tau) \sum_{k \geq 0} S_\tau(k) \right] \tag{19}
\end{align}
$$

This proves the first equality in the lemma. Also, we have:

$$
\begin{align}
S_\tau(k) &= \mathbb{P}(L > \tau + k|\mathcal{X}_\tau, L \geq \tau) \\
&= [1 - \mathbb{P}(L = \tau|\mathcal{X}_\tau, L \geq \tau)][(1 - \mathbb{P}(L = \tau + 1|\mathcal{X}_\tau, L \geq \tau)] \cdots [(1 - \mathbb{P}(L = \tau + k|\mathcal{X}_\tau, L \geq \tau)] \\
&= \prod_{m=0}^k (1 - h_\tau(m)) \tag{20}
\end{align}
$$

proving the second equality, and hence the lemma.

$\square$

**Lemma A.0.2.** $\mathbb{P}(L \leq \tau) = \mathbb{E}[(1 - \pi_\tau)h_\tau(0) + \pi_\tau]$

*Proof.* Using the law of iterated expectations:

$$
\begin{align}
\mathbb{P}(L \leq \tau) &= \mathbb{E}[\mathbb{1}(L \leq \tau)] = \mathbb{E}\left[\mathbb{E}\left[\mathbb{1}(L \leq \tau)|\mathcal{X}_\tau\right]\right] \tag{21} \\
&= \mathbb{E}[(1 - \pi_\tau)\mathbb{P}(L \leq \tau|\mathcal{X}_\tau, L \geq \tau) + \pi_\tau \mathbb{P}(L \leq \tau|\mathcal{X}_\tau, L < \tau)] \tag{22} \\
&= \mathbb{E}[(1 - \pi_\tau)\mathbb{P}(L = \tau|\mathcal{X}_\tau, L \geq \tau) + \pi_\tau] \tag{23} \\
&= \mathbb{E}[(1 - \pi_\tau)h_\tau(0) + \pi_\tau] \tag{24}
\end{align}
$$

$\square$

Now we are ready to prove Proposition 3.1:

Starting with the OTI problem and using Lemmas A.0.1 and Lemma A.0.2, while also noting that the critical event has a maximum duration $L_{max} < \infty$, so that $S_\tau(k) = 0, \forall k \geq L_{max}$, we have:

$$\tau^* = \arg\inf_\tau \mathbb{E}\left[(1 - \pi_\tau)\sum_{k=0}^{L_{max}-1}\prod_{m=0}^{k}(1 - h_\tau(m)) + (1 - \pi_\tau)C_\alpha h_\tau(0) + \pi_\tau C_\alpha\right] \quad (25)$$

$$= \arg\inf_\tau \mathbb{E}\left[(1 - \pi_\tau)R^H(\mathbf{H}_\tau) + \pi_\tau C_\alpha\right] \quad (26)$$

where

$$R^H(\mathbf{H}_\tau) := \sum_{k=0}^{L_{max}-1}\prod_{m=0}^{k}(1 - h_\tau(m)) + C_\alpha h_\tau(0) \quad (27)$$

with $\mathbf{H}_\tau = [h_\tau(0), h_\tau(1)\cdots h_\tau(L_{max} - 1)]^T$. This completes the proof of Proposition 3.1

## A.2 Proof of Theorem 4.1

The proof of this Theorem relies on the following key results from optimal stopping theory[9, 4, 11, 27] stated below.

**Definition A.1** (Optimal stopping problem with Markov representation). An optimal stopping problem $\tau^* = \arg\inf_\tau \mathbb{E}[\tilde{R}_\tau]$, is said to have a Markov representation if the risk process can be expressed as $\tilde{R}_j = g_j(\mathbf{X}_j)$ where the $g_j$ are a sequence of functions and $\{\mathbf{X}_j\}$ is a Markov process in the sense: $\mathbb{P}(\mathbf{X}_{j+1}|\mathcal{X}_j) = \mathbb{P}(\mathbf{X}_{j+1}|\mathbf{X}_j), \forall j$.

**Theorem A.1** (Optimal stopping policy for problems with Markov representation). *Assume that we have an optimal stopping problem with Markov representation, with associated risk process $\tilde{R}_j = g_j(\mathbf{X}_j)$. The optimal stopping policy $\varphi^*$ and corresponding optimal stopping time $\tau^*$ are given by:*

$$\tau^* = \inf\{j \geq 1 : \varphi_j(\mathcal{X}_j) = 1\}, \quad \varphi_j^*(\mathcal{X}_j) = \mathbb{1}(\tilde{R}_j \leq \tilde{V}_j(\mathcal{X}_j)) \quad (28)$$

*where $\tilde{V}_j(\mathcal{X}_j) = \text{ess}\inf_{\tau > j}\mathbb{E}[\tilde{R}_\tau|\mathbf{X}_j]$.*[4]

Now we proceed to prove Theorem 4.1:

First, we show that the OTI problem for Markov hazard rate processes is an optimal stopping problem with Markov representation.

**Lemma A.1.1.** *Given a Markov hazard rate process $\{(\mathcal{X}_j, \mathbf{H}_j)\}$ with associated risk process $\{\tilde{R}_j\} = \{R(\pi_j, \mathbf{H}_j)\}$, the optimal stopping problem $\tau^* = \arg\inf_\tau \mathbb{E}[\tilde{R}_\tau]$ has a Markov representation.*

*Proof.* The process $\{\mathbf{X}_j\}$ is a Markov process by definition. Further, the risk process may be written as $\tilde{R}_j = R(\pi_j, \mathbf{H}_j) = g(\mathbf{X}_j)$, since we have $\mathbf{H}_j = f_j(\mathbf{X}_j)$ and $\pi_j = \mathbb{1}(L < j|\mathbf{X}_j) = e_j(\mathbf{X}_j)$ where $e_j(\cdot)$ and $f_j(\cdot)$ are functions of $\mathbf{X}_j$. Thus, a Markov hazard rate process meets both conditions for an optimal stopping problem with a Markov representation. $\square$

Thus the consequences of Theorem A.1 hold. The optimal policy $\varphi^*$ calls for stopping at:

$$\tau^* = \min\{1 \leq j < L_{max} : \varphi_j^*(\mathcal{X}_j) = 1\}, \quad \varphi_j^*(\mathcal{X}_j) = \mathbb{1}(\tilde{R}_j \leq \tilde{V}_j(\mathcal{X}_j)) \quad (29)$$

where,

$$\tilde{V}_j(\mathcal{X}_j) = \text{ess}\inf_{\tau > j}\mathbb{E}[\tilde{R}_\tau|\mathbf{X}_j] = \min_{0 < l < L_{max} - j}\mathbb{E}[\tilde{R}_{j+l}|\mathbf{X}_j] \quad (30)$$

We either have $\pi_j = 0$ or $\pi_j = 1$. First, we consider the case when $\pi_j = 0$, so $\mathbb{1}(L \geq j|\mathbf{X}_j) = 1$:

---

[4]Note that the essential infimum (ess inf) of a set of random variables $\mathcal{X}$ is any extended random variable $Z$ having the properties [27]
(i) $\mathbb{P}(Z \leq X) = 1, \forall X \in \mathcal{X}$, and
(ii) $\{\mathbb{P}(Y \leq X) = 1, \forall X \in \mathcal{X}\} \implies \mathbb{P}(Y \leq Z) = 1, \forall X \in \mathcal{X}$.

For $l = 0$, $\mathbb{E}[\tilde{R}_j|\mathbf{X}_j]$ is the expected risk of stopping at time step $\tau = j$. If the critical event does not occur at time-step $j$, then the risk incurred by stopping at time-step $j$ is the expected time to event conditioned on $L > j$. Alternatively, if the critical event does occur at time-step $j$, stopping effectively misses the event, accruing risk of $C_\alpha$. Since the two cases occur with probabilities $(1 - h_j(0))$ and $h_j(0)$ respectively, we have:

$$\mathbb{E}[\tilde{R}_j|\mathbf{X}_j] = (1 - h_j(0)) \sum_{k=0}^{L_{max}-1} S_j(k|L > j) + C_\alpha h_j(0) \tag{31}$$

Since $S_j(k|L > j) = \frac{S_j(k)}{S_j(0)} = \frac{S_j(k)}{(1-h_j(0))}$:

$$\mathbb{E}[R_j|\mathbf{X}_j] = \sum_{k=0}^{L_{max}-1} S_j(k) + C_\alpha h_j(0) = \sum_{k=0}^{L_{max}-1} \prod_{m=0}^{k} (1 - h_j(m)) + C_\alpha h_j(0) \tag{32}$$

For $l = 1$, $\mathbb{E}[\tilde{R}_{j+1}|\mathbf{X}_j]$ is the expected risk of stopping at time step $\tau = j + 1$. If the critical event does not occur at time $j$ and $j + 1$, then the risk incurred by stopping at $j$ is the expected time to event conditioned on $L > j + 1$. Alternatively, if the critical event does occur at either time $j$ or $j + 1$, a policy that contemplates stopping at $j + 1$ effectively misses the event, accruing risk of $C_\alpha$. Since the two cases occur with probabilities $(1 - h_j(0))(1 - h_j(1))$ and $[h_j(0) + (1 - h_j(0))h_j(1)]$ respectively, we have:

$$\mathbb{E}[\tilde{R}_{j+1}|\mathbf{X}_j] = (1 - h_j(0))(1 - h_j(1)) \sum_{k=0}^{L_{max}-2} S_j(k+1|L > j+1) + C_\alpha[h_j(0) + (1 - h_j(0))h_j(1)] \tag{33}$$

Since $S_j(k + 1|L > j + 1) = \frac{S_j(k+1)}{S_j(1)} = \frac{S_j(k+1)}{(1-h_j(0)(1-h_j(1)))}$:

$$\mathbb{E}[\tilde{R}_{j+1}|\mathbf{X}_j] = \sum_{k=0}^{L_{max}-2} S_j(k + 1) + C_\alpha[h_j(0) + (1 - h_j(0))h_j(1)] \tag{34}$$

$$= \sum_{k=0}^{L_{max}-2} \prod_{m=0}^{k+1} (1 - h_j(m)) + C_\alpha[h_j(0) + (1 - h_j(0))h_j(1)] \tag{35}$$

Extending this approach we may write the expected risk of stopping $l$ steps ahead of the current step as:

$$\mathbb{E}[\tilde{R}_{j+l}|\mathbf{X}_j] = \sum_{k=0}^{L_{max}-l-1} S_j(k + l) + C_\alpha h_j(0) + C_\alpha \sum_{m=1}^{l} \left[ \prod_{k=0}^{m-1} (1 - h_j(k)) \right] h_j(m) \tag{36}$$

$$= \sum_{k=0}^{L_{max}-l-1} \prod_{m=0}^{k+l} (1 - h_j(m)) + C_\alpha h_j(0) + C_\alpha \sum_{m=1}^{l} \left[ \prod_{k=0}^{m-1} (1 - h_j(k)) \right] h_j(m)$$

By equation (30) the continuation value function $\tilde{V}_j(\mathcal{X}_j)$ for the optimal stopping rule is:

$$\min_{0<l<L_{max}-j} \left\{ \sum_{k=0}^{L_{max}-l-1} \prod_{m=0}^{k+l} (1 - h_j(m)) + C_\alpha h_j(0) + C_\alpha \sum_{m=1}^{l} \left[ \prod_{k=0}^{m-1} (1 - h_j(k)) \right] h_j(m) \right\}$$

$$:= V_j(\mathbf{H}_j) \tag{37}$$

Next, we consider the case when $\pi_j = 1$, so $\mathbb{1}(L < j|\mathbf{X}_j) = 1$:

In this case, $\tilde{R}_j = C_\alpha$. Also, $\mathbb{E}[\tilde{R}_{j+l}|\mathbf{X}_j] = C_\alpha$, $\forall l > 0$. Therefore, $\tilde{V}_j(\mathcal{X}_j) = C_\alpha$, so it is optimal to trigger intervention, setting $\varphi_j^*(\mathcal{X}_j) = 1$.

Finally, we have $\varphi_{L_{max}-1}^*(\mathcal{X}_j) = 1$, by definition, giving rise to the final form of the optimal policy.

This completes the proof of Theorem 4.1

## A.3 Proof of Corollary 4.1.1

Note that when $\pi_j = 0$, the immediate stopping risk for time $j$ can be written as:

$$\tilde{R}_j = \sum_{k=0}^{L_{max}-1} S_j(k) + C_\alpha h_j(0) \tag{38}$$

$$= T_j + C_\alpha h_j(0) \tag{39}$$

where $T_j$ is the expected time to event having observed $\mathcal{X}_j$. We note from equation (37) that the term $h_j(0)C_\alpha$ is also always present in the continuation value function $V_j(\mathbf{H}_j)$. This allows us to cancel this term on both sides of the inequality defining the optimal stopping policy, resulting in the simplified rule written in terms of $T_j$ as:

$$\varphi_j^*(\mathcal{X}_j) = \begin{cases} \mathbb{1}(T_j \leq V_j^{'}(\mathbf{H}_j)), & \text{if } j < L_{max} - 1 \text{ and } \pi_j = 0 \\ 1, & \text{else} \end{cases} \tag{40}$$

where,

$$V_j^{'}(\mathbf{H}_j) = \min_{0 < l < L_{max}-j} \left\{ \sum_{k=0}^{L_{max}-l-1} \prod_{m=0}^{k+l} (1 - h_j(m)) + \sum_{m=1}^{l} \left[ \prod_{k=0}^{m-1} (1 - h_j(k))h_j(m) \right] C_\alpha \right\} \tag{41}$$

This completes the proof of Corollary 4.1.1

# B  Model hyper-parameter settings

Table 1: model hyper-parameters

| Hyper-parameter | Description | value |
|---|---|---|
| encoder look-back | encoder input time-steps | 128 |
| batch size | 512 | 512 |
| learning rate (WBI) | model selected from $\{0.1, 0.01, 0.001, 0.0001\}$ | 0.001 |
| learning rate (OTI, TTE) | model selected from $\{0.1, 0.01, 0.001, 0.0001\}$ | 0.01 |
| epochs | early stopping on validation loss | patience: 10 |
| batches per epoch | training batches for one epoch | 100 |
| WBI window ($W$) | selected from $\{8, 16, 32, 64\}$ | 16 |
| encoder/decoder hidden units | | 16 |
| WBI intervention threshold | tuned in range $[0, 1]$ | step size: 0.001 |
| TTE intervention threshold | tuned in range $[0, 1]$. Threshold applied to $\frac{\hat{T}_j}{L_{max}}$ | step size: 0.001 |
| Decoder max steps (training) M | steps before censoring kicks in | 64 |
| $\lambda$ | for DDRSA loss | 0.75 |
| optimizer | | Adam |

# C  Experiments on synthetic data

In this section we further validate the OTI policy by constructing synthetic data for which the optimal policy appears to be a static threshold. We show empirically that the threshold function of the OTI policy does indeed approach static thresholds for this problem.

**Synthetic failure data:** A dataset of 512 series of length 300 time steps each was prepared by selecting a random change point $10\%$ to $35\%$ from the end of the series. A Normal distribution with $\mu_f = 0.2, \sigma_f = 1$ was used to draw i.i.d. samples representing co-variates in the region after the change point (failure region) while a Normal distribution with mean $\mu_0 = 0, \sigma_0 = 0.8$ was used to draw i.i.d. samples representing normal behavior in the period before the change point. The start of each sequence is randomly chosen in the range $0\%$ to $50\%$ of max sequence length, from the beginning of the series.

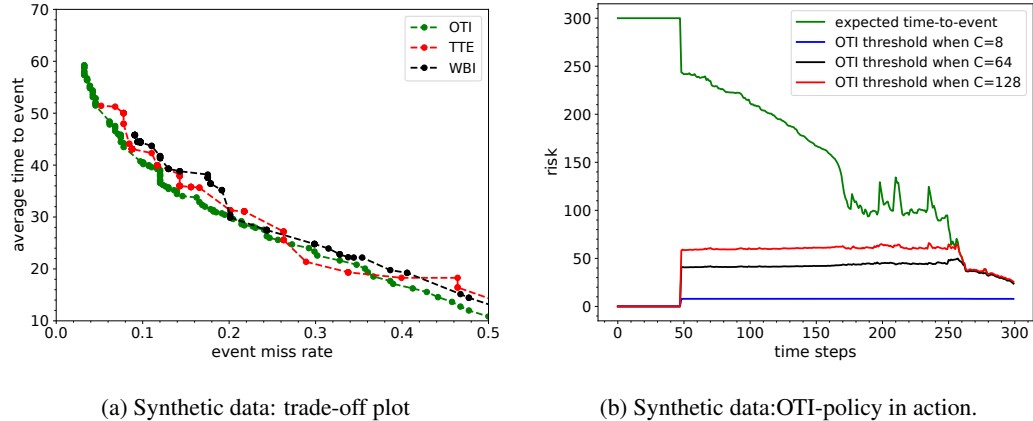

(a) Synthetic data: trade-off plot

(b) Synthetic data:OTI-policy in action.

Figure 4: Experiments on synthetic failure data.

Figure 4 shows the trade-off curves for this case. The results suggest that threshold policies seem to be optimal for this problem. We can see this empirically by observing the OTI policy on an example sequence. We see that the threshold function of the OTI policy does approach a constant threshold level. This experiment raises the question: Under what conditions are static threshold intervention policies optimal? This is a topic for future work.