# OpenReview forum: "When to Intervene: Learning Optimal Intervention Policies for Critical Events"
_NeurIPS.cc/2022/Conference — NeurIPS 2022 Accept_

### Official Review · Reviewer_vH1X · 2022-06-22

**Rating:** 5
**Confidence:** 2
**Soundness:** 4 excellent
**Presentation:** 3 good
**Contribution:** 3 good

**Summary:**

The paper formulates and studies the problem of optimally timed intervention (OPI) for the first time. More specifically, the problem is formulated as minimizing the expected time-to-event while keeping the probability of missing the critical event below a certain threshold. The authors show that the OTI problem is equivalent to an optimal stopping problem on the underlying survival process. Other theoretical characterizations are also performed showing that the optimal decision rule at each time step can be analytically determined by the conditional survival function and that the optimal policy is non-stationary in general (so static threshold policies are often sub-optimal). The authors also propose a dynamic deep recurrent survival analysis (DDRSA) architecture. Experimental results on two real-world datasets show that the proposed method provides a better tradeoff between the average time to event and event miss rate compared to some baselines.

**Questions:**

- Is the proposed OTI policy applicable only to the RNN model, or can it be more flexible? If it is more flexible, I thought it would be interesting to see if the proposed framework can work with a range of models to estimate the hazard rate function rather than focusing only on RNN.

- the OTI policy is expressed with the true hazard rate function for now. Would it be possible to theoretically characterize the accuracy of the function estimation and the performance of the resulting policy? In other words, how does the error of the function estimation propagate to the performance of the intervention policy?


**Limitations:**

I was unsure why it is necessary to focus only on the RNN model, even though the high-level OTI policy framework seems to be more flexible in my understanding. Then, it would be more comprehensive to show this flexibility in the experiment rather than focusing only on a single model.

**Strengths And Weaknesses:**

Strengths
- formulate the OTI problem and link it to both survival analysis and optimal stopping theory for the first time, deriving the analytical expression of the optimal intervention
- propose a method based on recurrent neural networks to lead an OTI policy
- promising empirical results of the proposed method compared to some minimum baselines
- the paper seems well written and the contributions were easy to follow


Weaknesses
- it seems that the experiment focuses only on RNN rather than trying to show the flexibility of the proposed framework. I could not see any particular reason why it is better to focus on RNN even though the proposed framework in section 3 does not necessarily rely on a particular model  (at least in my current understanding)

---

> ### Author Response · Authors · 2022-08-02
> **Response to Reviewer**
>
> Thank you for noting the key contributions of our work. We hope to have addressed your questions and mitigated perceived weaknesses.
>
> > Is the proposed OTI policy applicable only to the RNN model, or can it be more flexible? If it is more flexible, I thought it would be interesting to see if the proposed framework can work with a range of models to estimate the hazard rate function rather than focusing only on RNN.
>
> To calculate optimal interventions we just need to map covariate history to a conditional hazard rate function at each time index. The theory is flexible enough to support any model that can acheive this. For example, we could use the classic Cox proportional Hazards model with an parametric/empirically estimated baseline hazard over all time-steps and fit covariates (including time) over all time steps to produce a conditional hazard rate function at every time index.
>
> However, while it is true that one may adapt any method (in theory) that is used for Survival Analysis for this purpose, RNNs are a good fit for being able to model complex patterns in multi-variate time-series data and in summarizing observation history. Also, the DRSA RNN architecture by Ren et. al. has acheived state of the art performance relative to a number of alternatives including Kaplan Meier, Lasso-Cox, Survival random forests, Multi-Task survival analysis (MTLSA) and non-RNN deep learning models like DeepSurv and DeepHit [29].
>
> > the OTI policy is expressed with the true hazard rate function for now. Would it be possible to theoretically characterize the accuracy of the function estimation and the performance of the resulting policy? In other words, how does the error of the function estimation propagate to the performance of the intervention policy?
>
> This is a great research direction to pursue as a next step to this work. Note however, that we can (and are) empirically characterizing model fit by test set loss. We are also able to quantify the resulting policy performance over a range of  miss-rates. However, further theory may provide useful sample bounds, for example.

---

### Official Review · Reviewer_KFJC · 2022-07-07

**Rating:** 5
**Confidence:** 2
**Soundness:** 2 fair
**Presentation:** 2 fair
**Contribution:** 2 fair

**Summary:**

This paper studies the problem of triggering stopping interventions in dynamical systems before a critical event occurs, named the optimally timed intervention problem (OTI). The authors formulate this problem as that of minimizing the expected time to event (remaining time lost when the intervention is triggered before the critical event), while also minimizing the expected cost of an event happening (fixed cost when no intervention is triggered before the critical event). They show that two popular approaches, window-based intervention (WBI) and time-to-event (TTE), are sub-optimal for OTI, and they propose a new approach similar to TTE but with dynamic thresholds, which they show is optimal. They conduct experiments on two datasets which also support the superiority of their approach.

**Questions:**

I would like the authors to clarify the following points:
 - how does your derivation of the optimal intervention $\tau^\star$ translate into an optimal policy ?
 - what is a stationary process ? A stationary policy ?
 - given that any Markov process can be made homogeneous by including the index $j$ into each observation $X_j$, what is the point of considering non-homogeneous processes ?

Detailed comments:

p1 l21: networks(RNNs) -> typo

p2 l38-44: This whole description is a bit vague. "The critical event occurs" -> So there can only be one such event ? "The sequence of observations terminates after the occurrence of the critical event" -> what if there is no critical event ? Is there always a critical event ? If proper interventions happen at the right time, can't the system be kept alive for ever ? It seems what you describe does not allow that. "An intervention is considered late [...] if no intervention occurs" -> you say a late intervention is when there is no intervention... This sentence is contradictory by itself. I'd suggest to rework and clarify your setup description.

p2 l47: $\mathcal{X}_j \to \\{0,1\\}$ -> this notation is inconsistent. Earlier you define $\mathcal{X}_j$ as a single point in $\mathbb{R}^{j \times d}$. Now you use the same notation to depict the input space of a function.

p2 l53: again, the math here is quite loose and ambiguous. The index $j$ is defined twice in the left-hand-side and right-hand-side of the equation. On the left $j$ is given and constant, on the right $j$ is a free variable in a minimization problem. Are $\mathcal{X}_j$ on the left and on the right the same objects ? If not, how can $\tau$ be a function of $\mathcal{X}_j$ ? Also, what about the case where there is no intervention ? If the process reaches a terminal state $X_L$ without any intervention, then which value does the policy stopping time take ? Finally, the random variable $\tau$ does not depend only on a history at time $j$, it depends on the entire realization of the stochastic process and the policy $\phi$. Expressing $\tau$ as a function of $\mathcal{X}_j$ is really misleading.

p2 l58-60: This definition of OTI seem arbitrary. A motivation is missing, as to why such an objective makes sense in practice, or a reference to the literature if this objective is commonly accepted.

p2 eq1: Again, this equation is ill-defined. First, what is $L_\textit{max}$ ? What is $(L-\tau)^+$ ? These are not defined. Second, why isn't this  expression conditioned on the history $\mathcal{X}_j$ ? As it is written, equation (1) suggests that you are looking for a universal optimal time to intervene, $\tau^\star$, regardless of the current realization of the process. Surely this is sub-optimal, and at any time step $j$ the decision to intervene or not should be conditioned on the history $\mathcal{X}_j$, which can reduce the uncertainty about $L$.

p2 l67:  can be directly be -> typo

p3 l120: "system evolution towards a critical event is a non-stationary in general" -> I hardly get the meaning of that statement. Do you mean that your training or testing data are not i.i.d. realizations of random variables ($X_j$, $L$) ? Do you mean that each random variable $X_j$ follows a different distribution ? Is stationarity equivalent to homogeneousity ? A proper definition of stationarity is missing here.

p4 eq2: According to your setup description in Section 1.1, the existence of a history $\mathcal{X}_j$ implies that the process is not terminated yet, and hence $L>j$ is always true (or $L \geq j$, that is ambiguous from the definition of $L$). Thus I do not understand why the survival function is conditioned on $L>j$ in addition to $\mathcal{X}_j$ (or why the hazard rate is conditioned on $K \geq j$), as it does not bring any additional information.

p4 l160: From eq (2) I even have $S_j(0)=1$.

p4 l164: Again, $L_\textit{max}$ has not been defined yet.

p4 prop2.1: It seems that the notions of survival process and hazard rate process can be used interchangeably to define your OTI problem. I'd suggest to choose and stick to one of the two notions, this would simplify the manuscript a lot.

p5 eq5: I think the definition is missing $\forall j$. Otherwise this statement does not make much sense.

p5 l186: Is homogeneousity your definition of stationarity ? Furthermore, it seems that any Markovian survival process can be made homogeneous trivially, by including the index $j$ into each observed random variable $X_j$. What is the point of considering non-homogeneous Markov processes then ?

p5 l189: risk process risk process -> typo

p5 eq6: It is unclear here how this random variable $\tau^\star$ is related to a decision rule. At each time step $j$, a decision rule has to decide whether to stop or not, given only information from the past ($\mathcal{X}_j$). In equation (6), the optimal stopping time depends on the entire realization of the process (every $j$ has to be considered, in the right-hand-side of the equation). A clarification is missing, as to how equation (6) translates into a proper decision rule.

p5 eq7: $\prod_{m=0}^k (1-h_j(l+k))$ -> the $m$ index does not appear inside the factors here. This product does not make sense.

p5 l193: the optimal policy is stationary -> what does it mean for a policy to be stationary ?

p6 l239: $\mathbf{H}_0)$ -> typo

p6 l241: "produce conditional hazard rate" -> there is a limitation here to how many hazard rates your model can output, which directly depends on the model architecture and in particular the width of the final dense layer. This limitation is not discussed.

p6 l241: "a novel sequential loss" -> a description of this novel loss function is missing.

p8 fig2: dynamicaly -> typo

p8 l287: If in the end you model everything with hazard rates, which can be equivalently represented as survival functions, then I really suggest that you use only one of the two concepts in the paper. I do not see the additional value of expressing all your results using both concepts each time. This really hinders the readability of the paper.

p8 l292: If the OTI policy can be equivalently expressed as an adaptive threshold on the estimated time-to-event, then why bothering introducing hazard rates and survival functions in the first place ? This make me feel like the presented theoretical results are stated in an overly complicated fashion.

p9 fig3: These figures are missing information about the variance, such as boxplots or standard deviation curves.

p9 l320: Allmodels -> typo

**Limitations:**

Limitations are discussed very briefly. I see several limitations that are missing in the discussion. How should the cost C be chosen in practice ? What is the affect of the size of the predicted hazard rate vector ? What is the computational complexity of the proposed approach at test time ?

### rebuttal update

I have been through the author's response to my comments. I appreciate that the authors have now substantially simplified the paper, notably by moving the analysis in terms of survival process (a somewhat duplicate of the analysis in terms of hazard rate process) to the appendix. I also note that the authors have fixed several mathematical typos and errors from the original manuscript that I had pointed to.

However, I have also been through the updated manuscript, and I still find the mathematical notations and definitions to lack somewhat loose and ambiguous. I could not check in detail the entire revised paper due to the amount of changes and the limited time, but I have concerns already after reading the first few pages:

 - p2 l41: but bounded time-step $L < L\_{max}$ in the series. -> typo ? but at a bounded time step ?

 - p2 l43: before the the policy stops -> typo. Moreover, I still find this whole definition rather vague. What does "the policy stops" mean ? How different is that from saying "before an intervention" ? Does the policy stop / intervention necessarily triggers the end of the time series ? After an intervention at time step $\\tau$, is there still a final observation $X\_{\\tau + 1}$ ? If $X\_{L+1}=0$ indicates the end of the time series, then is this specific value forbidden, in the domain of the random variables $X\_j$ ? Despite the changes made, this whole definition still appears rather sloppy to me, which is problematic for a theoretical paper.

 - p2 l44: we can take $\\tau=L+1$ -> At this point $\\tau$ has not been introduced yet. Also, why not considering that the policy stops immediately, with $\\tau=L$ ? This would seem to make more sense, given that you consider interventions at $\\tau=L$ too late by definition.

 - p2 l50: Definition 1.1 -> First a suggestion. Because $j$ is somehow embedded into $\\mathcal{X}\_j$ (it is the length of the history), why not simply defining your intervention policy as $\\varphi(\\mathcal{X}\_j)$, instead of a sequence of $j$-indexed functions ? This would simplify a lot an otherwise heavy notation. Second, I again find the math a bit sloppy here. Earlier you defined $\\mathcal{X}\_j$ as a random variable, but here in $\\{\\mathcal{X}\_j\\} \\to \\{0,1\\}$ you seem to use it as the input space of a mapping. I understand the intention, but again this is a rather loose notation. A random variable and its support are not the same thing.

 - p2 l53: this means that intervention triggers after the first trigger are ignored -> typo ?

Again, I am not an expert in the field of OTI, but at the very least I should be able to read the problem setup unambiguously. To me the overall technical quality of the paper remains below the NeurIPS threshold.

Taking all of that into account I am willing to increase my rating to 4, but my recommendation remains rejection.

**Strengths And Weaknesses:**

### Originality

The proposed approach seems original, although I am not very familiar with the OTI problem.

### Quality

While the paper appears quite technical, I found the technical quality rather poor. The mathematical definitions used in the paper are loose and ambiguous at times, which affects readability and also brings doubt about the rigor of the presented results. Some parts of the reasoning are missing, such as how the optimal intervention $\tau^\star$ (Theorem 3.1, Corollaries 3.1.1 and 3.1.2) translates into an actual policy, or what the authors mean by a stationary process or a stationary policy. The experimental setup seems ok, but the experimetns are lacking a bit in size (only 2 datasets, only 5 and 3 values respectively for the cost $C$ for each dataset), and in presentation (no error bars or standard deviation).

### Clarity

I found the paper hard to read, due to several ambiguities in the notations and definitions. I also found the presentation of the proposed approach unnecessarily heavy and redundant. The authors derive their method twice from two equivalent concepts: the hazard rate and the survival function. I believe the presentation could be greatly simplified. Finally, the paper is missing a conclusions section.

### Significance

The presented analysis and approach might be significant to the OTI community. Still, the technical quality and presentation of the results is too poor to correctly assess significance in my opinion.

---

> ### Author Response · Authors · 2022-08-02
> **Response to reviewer**
>
> We thank the reviewer for noticing the originality of the work. To our knowledge this is the first formal formulation and study of the optimally timed intervention problem, combining survival analysis and optimal stopping theory.
>
> >how does $\tau^*$ translate into an optimal policy ?
>
> As given in Theorem 3.1, equation (6), page 5, the optimal policy (that gives rise to the optimal $\tau^*$ ) is  triggered when $R_{j}^H({\bf H}_j) \leq V_j({\bf H}_j)$.  The first such trigger  sets the stopping time and stops the policy. To make the separation of the optimal policy $\varphi^*$ from the optimal stopping time $\tau^*$ explicit we have revised as:
>
> $\tau^* = \min \\{1 \leq j \leq L_{max}: \varphi_j^*({\bf H}_j) = 1  \\}$ with  $\varphi_j^{*}({\bf H}_j) = \mathbb{1}(R_j^H({\bf H}_j) \leq V_j({\bf H}_j))$
>
>
> > what is a stationary process ? A stationary policy ?
>
> **Stationary Process:** A  stochastic process whose unconditional joint probability distribution does not change when shifted in time. See for example the definition in the books "Introduction to Random Processes", by Eugene Wong (page 76)
>
> **Stationary Policy:** A policy is stationary if the decision rule at each epoch does not depend on the time-index. See for example the definition in the book "Markov Decision Processes", by Puterman (pages 21 and 22).
>
> In the revision we have removed entirely or used alternate language.
>
> > given that any Markov process can be made homogeneous by including the index $j$ into each observation $\mathcal{X}_j$, what is the point of considering non-homogeneous processes ?
>
> We respectfully disagree. To the best of our knowledge a non-homogeneous Markov process cannot be made homogeneous by including the time index into the state.
>
> ### Detailed Responses
>
> > So there can only be one such event?....
>
> Footnote 2 on page 2 covers this under case of policy reactivation which we say are treated as independent realizations of the stochastic process. So if we recover from the occurrence of the first event and re-activate the policy to continue to monitor for a second event, or re-activate the policy to continue monitoring after an intervention, this is a case of policy reactivation.
>
> > Is there always a critical event?
>
> When we say (on line 40), that "the critical event occurs", we are assuming that the event occurs eventually.
>
> > what if there is no critical event?
>
> We simply treat critical events that are unobserved as censored (supplement: p7 lines 126-130).
>
> > The index $j$ is defined twice
>
> The correct version should be $\tau := \min \\{ j \geq 1 :  \varphi_j({\mathcal X}_j) = 1 \\}$. This defines a valid stopping time.
>
> >  If the process reaches a terminal state $\mathcal{X}_L$  without any intervention, then which value does the policy stopping time take?
>
>  All we need to know is that the stopping time will occur after the critical event in this case, and this determination can be made at time step $L$. This corner case is now explicitly covered in the revision. We can take $\tau = L+1$.
>
> > the random variable $\tau$.. depends on the entire realization of the stochastic process and the policy $\varphi$.
>
> One can determine if $\tau=j$ or not by only considering ${\mathcal X}_j$. Clearly, this is possible from our definition $\tau := \min \\{ j \geq 1 :  \varphi_j({\mathcal X}_j) = 1 \\}$. The first time index where ${\mathcal X}_j$ triggers the decision rule, $\varphi_j({\mathcal X}_j) = 1$ sets $\tau=j$.
>
> > Why isn't this expression conditioned on the history $\mathcal{X}_j$?
>
> The conditioning is implicit. Note also, that the formulation of Proposition 2.1, equations (3) and (4) expose the dependence of the OTI objective on the co-variates, since $h_{\tau}(k)$ depend on the covariates.  We have added a proof sketch for Proposition 2.1 in the revised version, making this point explicit. The notation used in the formulation are used routinely in optimal stopping setups [See books by Poor and Shirayaev]. Note the implicit inclusion of $\mathcal{X}_j$ in all these formulations.
>
> >  there is a limitation here to how many hazard rates your model can output
>
> See supplement, page 8, Appendix B .
>
> > ...a description of this novel loss function is missing.
>
> See supplement, page 8, Appendix B.
>
> > .. why bothering introducing hazard rates and survival functions in the first place ?
>
> * First, expected time to event needs to be estimated at each time. We can train a model using for example MSE, but in that case we cannot handle cases where the observation of critical event is censored.
>
> * How does one tune the thresholds at each time step? There is a combinatorial explosion of possible threshold settings covering all time steps and we run the risk of serious overfitting. Our formulation  simply computes the optimal thresholds to use at each time step once hazard rates are estimated. This is powerful.
>
> >  information about the variance
>
> In supplement, page 8 Appendix C, Table 2.

---

> ### Author Response · Authors · 2022-08-02
> **More elaborate responses**
>
> We thank the reviewer for noticing the originality of the work. To our knowledge this is the first formal formulation and study of the optimally timed intervention problem, combining survival analyis and optimal stopping theory.
>
> >how does your derivation of the optimal intervention $\tau^*$ translate into an optimal policy ?
>
> As given in Theorem 3.1, equation (6), page 5, the optimal policy that gives rise to the optimal $\tau^*$  is  triggered when $R_{j}^H({\bf H}_j) \leq V_j({\bf H}_j)$.  The first such trigger (note the min operation in the equation for stopping time) sets the stopping time to the corresponding time index and stops the policy. To make the separation of the optimal policy $\varphi^*$ from the optimal stopping time $\tau^*$ explicit we have revised as:
>
> $\tau^* = \min \\{1 \leq j \leq L_{max}: \varphi_j^*({\bf H}_j) = 1  \\}$ with  $\varphi_j^{*}({\bf H}_j) = \mathbb{1}(R_j^H({\bf H}_j) \leq V_j({\bf H}_j))$
>
>
> > what is a stationary process ? A stationary policy ?
>
> We just use the following standard well-known definitions:
>
> **Stationary Process:** A stationary process is a stochastic process whose unconditional joint probability distribution does not change when shifted in time. See for example the definitions in the books "Introduction to Random Processes", by Eugene Wong (page 76) or the book "Stochastic Processes", by Robert G. Gallagher (page 127).
>
> **Stationary Policy:** A policy is stationary if the decision rule at each epoch does not depend on the time-index. See for example the definition in the book "Markov Decision Processes", by Martin L. Puterman (pages 21 and 22).
>
> **Stationary Optimal Stopping Problem:** With regard to optimal stopping problems, any optimal stopping problem that has an underlying Homogeneous Markov Process and a time-invariant reward function is also said to be stationary [See page 41, in the book Quickest Detection by H. Vincent Poor and Olympia Hadjiliadis]
>
> Although these are standard definitions, to avoid potential confusion, in the revision we have removed these terms whenever an alternative (such as time-invariant) can be used or where it was included for completeness, but not central to the results (ex: mentions to Homogeneous process and Stationary problems were removed).
>
>
> > given that any Markov process can be made homogeneous by including the index $j$ into each observation $\mathcal{X}_j$, what is the point of considering non-homogeneous processes ?
>
> We respectfully disagree. To the best of our knowledge a non-homogeneous Markov process cannot be made homogeneous by including the time index into the state.  A Markov process is termed homogeneous if $\mathbb{P}(X_{j+1}|X_j)$ does not depend on $j$, $\forall j$. Adding $j$ into the state vectors does nothing to change the distribution of the other random components of $X_j$. So if that distribution shifts with time the process will still be non-homogeneous.
>
> > "The critical event occurs" -> So there can only be one such event ? If proper interventions happen at the right time, can't the system be kept alive for ever ? It seems what you describe does not allow that.
>
> As we have stated (p1 line 41) the sequence of observations terminates after the observance of the critical event. This forms a single realization of the stochastic process. Also, as stated (p2 line 52) all interventions are assumed terminal, also stopping the observations. Footnote 2 on page 2 covers the case of policy reactivation which we say are treated as independent realizations of the stochastic process. So if we recover from the occurrence of the first event and re-activate the policy to continue to monitor for a second event, or re-activate the policy to continue monitoring after an intervention, both these cases will be considered as an independent set of process observations resulting from policy reactivation. We have called out these special cases more explicitly in the revision.
>
> > Is there always a critical event?
>
> When we say (on line 40), that "the critical event occurs at a random time-step $L$ in the series", we are simply assuming that the event occurs eventually. We have made this more explicit by using the language: "the critical event is assumed to occur..." instead of "the critical event occurs..."
>
> > what if there is no critical event?
>
> As specifically called out in the supplemental material (supplement: p7 lines 126-130) this assumption does not pose a serious limitation in practice thanks to our survival analyis formulation. While fitting the survival model we simply treat critical events that are unobserved as censored. We have moved discussion of this case from supplement to the main paper.

---

> > ### Comment · Reviewer_KFJC · 2022-08-04
> > **About non-homogeneous processes**
> >
> > Thank you for these responses. I would like to clarify on one point.
> >
> > >> given that any Markov process can be made homogeneous by including the index $j$ into each observation $X_j$, what is the point of considering non-homogeneous processes ?
> > >
> > > We respectfully disagree. To the best of our knowledge a non-homogeneous Markov process cannot be made homogeneous by including the time index into the state. A Markov process is termed homogeneous if $p(X_{j+1} | X\_{j})$ does not depend on $j$, $\\forall j$. Adding $j$ into the state vectors does nothing to change the distribution of the other random components of $X_{j}$. So if that distribution shifts with time the process will still be non-homogeneous.
> >
> > Let me try to convince you. Say that you include $j$ in the process state, we now have a new state $\\hat{X}\_j = (X_{j}, j)$. Then, we have a new process with transition probabilities $p(\\hat{X}\_{j+1} | \\hat{X}\_{j}) = p(\\hat{X}\_{j+1} | X\_{j}, j)$. This new transition probability does not depend on $j$ any more, in the sense that $p(\\hat{X}\_{j+1} | \\hat{X}\_{j}, j) = p(\\hat{X}\_{j+1} | X\_{j}, j, j) = p(\\hat{X}\_{j+1} | X\_{j}, j) = p(\\hat{X}\_{j+1} | \\hat{X}\_{j})$, or equivalently $\\hat{X}\_{j+1} \\perp j \\mid \\hat{X}\_{j}$. Whichever information the time index $j$ could bring about the next state $\\hat{X}\_{j+1}$ is already present in the current state $\\hat{X}\_{j}$.
> >
> > > Adding $j$ into the state vectors does nothing to change the distribution of the other random components of $X_{j}$.
> >
> > First, unless your process is stationary (in the sense $p(X\_j) = p(X\_k)$, $\\forall j, k$), then knowledge of $j$ does change the distribution of $X_{j}$, i.e., $p(X\_j)$ changes for different values of $j$. If your process is non-homogeneous, then clearly the distribution $p(X\_{j+1} | X\_{j})$ also changes for different values of $j$, by definition.
> >
> > > So if that distribution shifts with time the process will still be non-homogeneous.
> >
> > With the time index-augmented states, $p(\\hat{X}\_{j+1} | \\hat{X}\_{j})$ does not change for different values of $j$, because $\\hat{X}\_{j+1} \\perp j \\mid \\hat{X}\_{j}$. There can be no distribution shift any more, so the process must be homogeneous.

---

> > > ### Author Response · Authors · 2022-08-05
> > > **True, but not relevant**
> > >
> > > Thank you for your clarification of this point. It makes sense. However, **why is it relevant?**
> > >
> > > All we have assumed is that the process in Markovian, so why do we need to assume that it is Markovian and Homogeneus (with time embedded in the state). It brings nothing anyway if the two are equivalent.
> > >
> > > Specifically,  **this does nothing to diminish the results** of  Theorem 3.1 or 3.1.1! The optimal policy is still a time-varying threshold on expected time to event, and the analytical form of the optimal policy would not change.

---

> ### Author Response · Authors · 2022-08-02
> **More elaborate responses-2**
>
> > "An intervention is considered late [...] if no intervention occurs" -> you say a late intervention is when there is no intervention... This sentence is contradictory by itself.
>
> When we say that we consider an intervention late if no intervention occurs at any time-step $j<L$, what we are saying is that we consider interventions at $j=L$ and beyond (since the critical event itself requires an intervention) too late by definition. We have added this is the revision.
>
>
> >  $\mathcal{X}_j \rightarrow \{0, 1\}$  this notation is inconsistent.
>
> Typo. Should be $\\{\mathcal{X}_j\\} \rightarrow \\{0, 1\\}$
>
> > The index $j$ is defined twice
>
> The correct version should be $\tau := \min \\{ j \geq 1 :  \varphi_j({\mathcal X}_j) = 1 \\}$. This defines a valid stopping time (see for example the same definition used on page 137 in the Chapter on sequential detection in the book: An Introduction to Signal Detection and Estimation, by H. Vincent Poor).
>
> > what about the case where there is no intervention? If the process reaches a terminal state $\mathcal{X}_L$  without any intervention, then which value does the policy stopping time take?
>
> In the case when the critical event occurs before the the policy stops, we are in effect forcing the policy to stop at the very next time-step, so we can take $\tau = L+1$. In this case we can assume a final observation $X_{L+1} = {\bf 0}$ denoting that the critical event has occured, that always stops the policy. This is for technical completeness only since we never actually use this in the derivations. All we need to know is that the stopping time will occur after the critical event in this case, and this determination can be made at time step $L$. This corner case is now explicitly covered in the revision.
>
> > the random variable $\tau$ does not depend only on a history at time $j$.
>
>  One can determine if $\tau=j$ or not by only considering ${\mathcal X}_j$. Clearly, this is possible from our definition $\tau := \min \\{ j \geq 1 :  \varphi_j({\mathcal X}_j) = 1 \\}$. The first time index where ${\mathcal X}_j$ triggers the decision rule, $\varphi_j({\mathcal X}_j) = 1$ sets $\tau=j$. In fact, as stated in footnote 2 on page 2, this is property defines a stopping time RV (see the book "Stochastic Processes", by Robert G. Gallagher, pg 234 or the book Quickest Detection by H. Vincent Poor and Olympia Hadjiliadis, pg 24)
>
> > This definition of OTI seem arbitrary. A motivation is missing, as to why such an objective makes sense in practice, or a reference to the literature if this objective is commonly accepted.
>
> We have added motivation and references to literature in the revision.
>
> The expected time-to-event, also called expected residual life or mean residual life (MRL) is a quantity that is well studied in literature with application in multiple domains [chapter 4 of Chin-Diew Lai, Min Xie, Stochastic Aging and Dependence for Reliability](\url{https://link.springer.com/chapter/10.1007/0-387-34232-X_4?noAccess=true}).
>
> A related objective also appears in optimal stopping formulations for Bayesian quickest change point detection problem [See page 103, equation (5.3) in the book Quickest Detection by H. Vincent Poor and Olympia Hadjiliadis or Chapter 2 page 13, equation (2.3) in the book Stochastic Disorder Problems by Albert Shirayaev], also called the Shirayaev problem, where the objective is to monitor a stochastic process for the quickest possible detection of a change point or stochastic disorder. Here the tradeoff is between stopping too early (proabability of a false alarm) and the expected time delay in detecting the change point, after it has occured.
>
> > What is $L_{max}$?:
>
> It is simply an upper bound on $L$. We have Included it in the OTI setup.
>
>
> > What is $(L-\tau)^{+}$:
>
> This follows from the context provided by preceding lines 58-60 (on page 2) where we say "minimizes expected time-to-event". We now say "minimizes expected time-to-event $E[ (L - \tau)^{+} ]$".

---

> ### Author Response · Authors · 2022-08-02
> **More elaborate responses-3**
>
> > Why isn't this expression conditioned on the history $\mathcal{X}_j$? As it is written, equation (1) suggests that you are looking for a universal optimal time to intervene, $\tau^*$, regardless of the current realization of the process. Surely this is sub-optimal, and at any time step  the decision to intervene or not should be conditioned on the history $\mathcal{X}_j$, which can reduce the uncertainty about $L$.:
>
> The conditioning is implicit. Note also, that the formulation of Proposition 2.1, equations (3) and (4) expose the dependence of the OTI objective on the co-variates, since $h_{\tau}(k)$ depends on the covariates. This was implicit in the original formulation.
>
> The conditioning on a specific $\mathcal{X}_j$ for a given $j$ alone would not be sufficient since we are looking for $\tau^*$ that will be optimal over all possible trajectories, so need to average over the trajectories. The proofs in the supplemental material make the dependence on observation history explicit. See for example Appendix A, Lemma A.1 and A.2 that use the law of iterated expectation to do this. Thus, we see that the outer expectation considers all possible observation trajectories that stop at $\tau$. Naturally optimizing over $\tau$ considers all possible stopping times as well. We have added a proof sketch for Proposition 2.1 in the revised version, making this point explicit.
>
> The notation used in the formulation and manipulations like the one we use above to create introduce dependence on $\mathcal{X}_j$ are used routinely in optimal stopping setups [See page 103, equation (5.3) in the book Quickest Detection by H. Vincent Poor and Olympia Hadjiliadis or Chapter 2 page 13, equation (2.3) in the book Stochastic Disorder Problems by Albert Shirayaev]. Note the implicit inclusion of $\mathcal{X}_j$ in all these formulations.
>
> > "system evolution towards a critical event is a non-stationary in general" -> I hardly get the meaning of that statement. Do you mean that your training or testing data are not i.i.d. realizations of random variables ($\mathcal{X}_j$, $L$) ? Do you mean that each random variable $\mathcal{X}_j$ follows a different distribution ? Is stationarity equivalent to homogeneousity? A proper definition of stationarity is missing here.
>
> Answered in our Response to key questions. However revised to remove mention of non-stationarity. We now say "system evolution towards a critical event is complex"
>
> > According to your setup description in Section 1.1, the existence of a history $\mathcal{X}_j$ implies that the process is not terminated yet, and hence $L > j$ is always true (or $L \geq j$, that is ambiguous from the definition of $L$). Thus I do not understand why the survival function is conditioned on  in addition to  (or why the hazard rate is conditioned on $L \geq j$), as it does not bring any additional information.
>
> There was a typo. The conditioning is on $L \geq j$, and not $L > j$. Note that the correct version appears in the proofs provided in the supplemental material. You are correct that observation of $\mathcal{X}_j$ implies $L \geq j$. The reason to keep it in the notation was to make the notation similar to survival analysis notation for conditional survival functions. See \url{https://www.pauldickman.com/software/stata/conditional-survival/}, while also highlighting conditioning on $\mathcal{X}_j$ which is not done in a traditional survival analysis where one does not take observations over time. Nevertheless, this is not a burden since we only deal with $h_j(k)$ and $S_j(k)$ which are compact.
>
> > From eq (2) I even have $S_j(0)=1$.
>
> After the typo fix above and correct conditioning on  $L \geq j$, we do not have this, only $S_j(-1)=1$.
>
> > It seems that the notions of survival process and hazard rate process can be used interchangeably to define your OTI problem. I'd suggest to choose and stick to one of the two notions, this would simplify the manuscript a lot.
>
> We had included this for completeness. We have revised by removing the survival function formulation from the paper, and keeping only the hazard rate formulation.
>
> > I think the definition is missing $\forall j$. Otherwise this statement does not make much sense.
>
> Indeed. We have added this.
>
> > $\prod_{m=0}^{k} (1 - h_j(l+k))$ -> the  index does not appear inside the factors here. This product does not make sense.
>
> This is clearly a typo. The $k$ in the hazard function arguement should be $m$. Note that the correct version appears in the proof provided in the supplemental material.

---

> ### Author Response · Authors · 2022-08-02
> **More elaborate responses-4**
>
> > "produce conditional hazard rate" -> there is a limitation here to how many hazard rates your model can output, which directly depends on the model architecture and in particular the width of the final dense layer. This limitation is not discussed.
>
> We have discussed this. See supplement, page 8, Appendix B for a discussion of this issue. Moved into the revised main paper.
>
> First, the model architecture is such that we can generate a long sequence of hazard rates (the decoder architecture (Figure 1 of Supplement, now in main paper) is time-distributed dense on top of a RNN, not a single dense layer with fixed width). During training time we may restrict the decoder time-steps to a fixed value ($\leq L_{max}$) and consider events outside this as censored. This is an advantage of using the survival formulation and allows such handling of long sequences.
>
> > "a novel sequential loss" -> a description of this novel loss function is missing.
>
> We have discusssed this. See supplement, page 8, Appendix B for a discussion on this.
>
> > If in the end you model everything with hazard rates, which can be equivalently represented as survival functions, then I really suggest that you use only one of the two concepts in the paper. I do not see the additional value of expressing all your results using both concepts each time. This really hinders the readability of the paper.
>
>  We had included this for completeness. We have revised by removing the survival function formulation from the paper, and keeping only the hazard rate formulation. The survival formulation is moved to the supplemental material.
>
> > If the OTI policy can be equivalently expressed as an adaptive threshold on the estimated time-to-event, then why bothering introducing hazard rates and survival functions in the first place ? This make me feel like the presented theoretical results are stated in an overly complicated fashion.
>
>  A key contribution of our work is the significant insight that the OTI policy can be expressed as an adaptive threshold on expected time to event. Regarding the need to model hazard rates once we know that the optimal policy is a time-varying threshold on the expected time-to-event:
>
>
> * First, expected time to event needs to be estimated at each time. We can train a model using for example MSE, but in that case we cannot handle cases where the observation of critical event is censored. The survival formulation allows for this quite elegantly.
>
> * Even if we take the estimation of expected time to event as a given, how does one tune the thresholds at each time step? There is a combinatorial explosion of possible threshold settings covering all time steps, and even if some search strategy is deviced we run the risk of serious overfitting on very small validation datasets (critical events are rare). Our formulation uses an analytical decision rule, so we can simply compute the optimal thresholds to use at each time step once hazard rates are estimated. This is a powerful capability.
>
>
>
> > These figures are missing information about the variance, such as boxplots or standard deviation curves.
>
> Please see Supplemental material, page 8 Section C, Table 2.

---

> ### Author Response · Authors · 2022-08-05
> **minor revision**
>
> There was a cut and paste issue with the prior revison (only in Section 1.1) which caused some sentences to be placed in above the defintions of the quantities they were refering to. We have made a very minor revision to the problem setup to fix this along with tightening some notation further.

---

> ### Author Response · Authors · 2022-08-05
> **Notation for decision rules**
>
> >p2 l50: Definition 1.1 -> First a suggestion. Because $j$  is somehow embedded into  ${\mathcal X}_j$  (it is the length of the history), why not simply defining your intervention policy as $\varphi({\mathcal X_j})$, instead of a sequence of j-indexed functions ?
>
> The domain of the function  $\varphi_j$  is $\mathbb{R}^{d \times j}$ while the domain of  $\varphi_k$ is $\mathbb{R}^{d \times k}$  which are different and so  $\varphi_j$ is not the same function as $\varphi_k$.
>
> We would request you to give the problem statement (which is entirely only a few lines  on Section 1 after all),  one more read, and reconsider your rating. We have tried to make the notation as tight as possible.

---

> > ### Comment · Reviewer_KFJC · 2022-08-08
> > **Comment**
> >
> > > Would you not consider this suggested notation "loose"? After all, the domain of the function $\varphi_j$ is $\mathbb{R}^{d \times j}$ while the domain of $\varphi_k$ is $\mathbb{R}^{d \times k}$ which are different and so $\varphi_j$ is not the same function as $\varphi_k$.
> >
> > Consider two disjoint sets $A$ and $B$, and two mappings $f_A: A \\to C$, and $f_B: B \\to C$. You can define a new set $D = A \cup B$, and merge your two mappings into a single one, $f: D \\to C$. This is not different from considering a single policy (mapping) which takes as input a history $\\mathcal{X} \in \\bigcup_{j=1}^{L_{max}} \\mathrm{R}^{d \times j}$. This is commonly done in POMDPs for example, where the optimal policy must consider a variable-length history of observations. See for example: Sufficient statistics in the optimum control of stochastic systems, C. Striebel, 1965
> >
> > > We would request you to give the problem statement (which is entirely only a few lines on Section 1 after all), one more read, and reconsider your rating.
> >
> > I would like to remind you that I am not an author of this paper, so I will not be writing your problem statement. I am acting as a reviewer voluntarily, using time that I could spend on my own research to read and give you an honest feedback on your work. You might like it or not, but I think I am being fair by giving you detailed comments and suggestions. You can decide to use it constructively or not. I understand how stressful the reviewing process can be as an author, but there is no need for being disrespectful to reviewers. It can only do you disservice.

---

> > > ### Author Response · Authors · 2022-08-09
> > > **Further clarifications**
> > >
> > > Dear reviewer,
> > >
> > > We do not intend any disrespect. Indeed, we deeply value your comments on our paper and your services to NeuRIPs.
> > >
> > > Thank you for clarifying what you meant by making the decision rules independent of $j$. It is certainly an interesting approach.
> > >
> > > However, indexing sequential decision rules by time index is well-accepted. (See for example the definition in the book "Markov Decision Processes", by Puterman (page 21), the classic book "An Introduction to Signal Detection and Estimation" by H. Vincent(page 137), or the book "Mathematical Statistics: A Decision Theoretic Approach", by Thomas Ferguson (page 310)) all use the same notation we do.
> > >
> > >
> > > In response to your review we have substantially re-written Section 1.1, which introduces the relevant notation, for further clarity. In the following we summarize the key points.
> > >
> > > > Is there always a critical event?
> > >
> > > Yes, by assumption in the theory.
> > >
> > > > what if there is no critical event?
> > >
> > > We can handle unobserved critical events in practice due to our survival formulation.
> > >
> > > >  If the process reaches a terminal state $\mathcal{X}_L$  without any intervention, then which value does the policy stopping time take?
> > >
> > > The optimal stopping time need not take any value when the critical event occurs. It is only defined to take a value when a decision rule fires (see definition 1.2).  We ensure that we have at least one intervention at or before $L_{max}-1$, so stopping time takes a value for every process realization. This is ensured by defining $\varphi_{L_{max}-1}({\mathcal X_{L_{max}-1}}) := 1$. In cases where the occurrence of the critical event makes it impossible to collect further observations beyond $L$ in practice, we can can simply fill such observations with dummy values (ex: zeros) without any loss of generality since these observations are not predictive of the critical event and hence do not feature in computing optimal intervention decisions (i.e. they are not used to estimate conditional hazard rate functions since these require future event occurrence).
> > >
> > > >  What does "the policy stops" mean ?
> > >
> > > once the first decision to intervene is triggered on a given realization of the monitored process, the policy effectively stops, i.e. an intervention action is assumed to be initiated and all future policy triggers for that process realization are ignored.
> > >
> > > >Does the policy stop / intervention necessarily triggers the end of the time series ?
> > >
> > > No. The theory allows observations after an intervention in the problem formulation. So an intervention policy can consider these covariates and fire multiple times, however, the objective function in the OTI problem is designed to consider only the first policy trigger.
> > >
> > > Note that we are given complete process realizations at least until the critical event. In formulating the optimal policy the considered interventions are only hypothetical, so do nothing to actually stop the series. Once the optimal policy is put into action (inference mode) then the first trigger can indeed stop the series via physical intervention, but this has no impact on the problem formulation or derivation of optimal policy.
> > >
> > >
> > > > If $X_{L+1}$ indicates the end of the time series, then is this specific value forbidden, in the domain of the random variables  ?
> > >
> > > No, such values are not forbidden, since co-variate values beyond event occurrence are never used to fit the hazard rate functions.

---

> ### Author Response · Authors · 2022-08-07
> **Rebuttal revision**
>
> Dear reviewer,
>
> We have updated the draft in incorporating your suggestions.  Specifically, we have addressed your comments on notation to the best of our ability.
>
> As the end of the discussion period is fast approaching, please do let us know if you feel other changes or clarifications are needed.

---

### Official Review · Reviewer_mWmk · 2022-07-11

**Rating:** 6
**Confidence:** 3
**Soundness:** 4 excellent
**Presentation:** 3 good
**Contribution:** 3 good

**Summary:**

This paper studies the problem of deciding when to intervene and “stop” a stochastic process before it transitions to a critical event. A new problem setting is defined called optimally timed intervention (OTI), where the goal is to minimize the expected time between intervention and the event L under the constraint that the probability of missing is small. An optimal policy is derived through the introduction of survival and hazard processes. The theory is then used to motivate an RNN approach to predict these processes. The method is evaluated on real-world data and appears to marginally outperform a baseline and significantly outperform a naive strategy.

**Questions:**

- Is it true that the original problem can be equivalently formulated as a standard optimal stopping problem (before even considering the survival and hazard processes)? In that case, what prevents one from directly using existing tools from optimal stopping and how would they compare with the optimal policy derived in this paper through the survival/hazard processes?
- Can the authors clarify the novelty of the survival and hazard processes? These seem to be presented as new ideas, but also the paper by Ren et al also seems to use the hazard process.


**Limitations:**

Limitations were addressed.

**Strengths And Weaknesses:**

Strengths
- This is not within my area so I cannot comment on the significance compared to related work, but the paper seems to do a good job differentiating itself.
- The OTI seems to be an interesting and well-motivated problem to study (again I can not really comment on novelty)
- The theory about computation of the optimal policies given the known survival/hazard rates is also fairly comprehensive.
- The experiments are promising, suggesting that the theory and insights developed in the paper have strong practical implications.

Weaknesses
- It doesn’t seem that it is ever described formally what the relationship is between the covariates of the process and the event time L. All that is known in the problem definition is that L is some random variable and a stopping time. Nevertheless in the experiments this is implicitly posed as a supervised learning problem where the covariates of the stochastic process are somehow predictive of L. While this makes intuitive sense and seems to be sufficient for the experiments, I think it would be helpful to make this more rigorous so that we can reason about familiar characteristics of statistical learning problems like realizability, noise, etc. In general the shift from the OTI theory to the experiments is a bit abrupt and I think it would have been helpful to explain in math more clearly how that jump is made. Some of these details are in the appendix already, but it would be helpful to bolster the formalization and have them in the main paper.
- It doesn’t seem that one can “ask” for a specific probability of failure (missing L) and then pay the price of this ask in the expected time to intervene, like one would expect to see in many similar learning/decision problems. Instead this is controlled implicitly by $C_{\alpha}$ which is a hyperparameter that must be set.
- I think that the experiments could be a lot more compelling if there were a plot showing the trade-off between the probability of failure and the expected time to intervene so that one could see visually what shape this trade-off curve takes (by setting different values of $C_{\alpha}$). Perhaps this is not feasible with the real-world data. However, even synthetic experiments could suffice to exhibit this.

---

> ### Author Response · Authors · 2022-08-02
> **Response to Reviewer**
>
> Thank you for noting that the theory and insights developed have strong practical implications. This was a key motivating factor for our work. We hope we have fully addressed your questions and mitigated perceived weaknesses to your satisfaction.
>
> > what prevents one from directly using existing tools from optimal stopping and how would they compare with the optimal policy derived in this paper through the survival/hazard processes?
>
> The application of optimal stopping is problem specific and involves art in manipulating the problem to a point where the fundamental Theorems of optimal stopping may be applied. To get an idea of this, please see the [notes by Tom Ferguson] (https://www.math.ucla.edu/~tom/Stopping/Contents.html). Each optimal stopping problem often requires a different thought process.
>
> Starting with the OTI problem formulation in Section 1.2, to apply optimal stopping theory to derive an optimal policy would typically require assumptions on the prior probability distribution of the critical event and either i.i.d or some distributional assumption on joint-covariate evolution. Also, it is very hard to use with more than a few co-variates in this setting.
>
> To see the type of solutions attempted in literature to solve optimal stopping problems in a different but related setting with a similar objectives see the optimal stopping formulations for Bayesian quickest change point detection  [the book Quickest Detection by H. Vincent Poor and Olympia Hadjiliadis or  the book Stochastic Disorder Problems by Albert Shirayaev].
>
> The significance of our novel survival formulation is that we are able to solve the optimal stopping problem in a fairly general setting for critical event interventions not requiring stringent distributional assumptions and also allowing for high-dimensional co-variates.
>
> > Can the authors clarify the novelty of the survival and hazard processes? ...the paper by Ren et al also seems to use the hazard process
>
> The primary contribution of our work is in deriving analytic optimal intervention policies for critical events from the observation of co-variates that evolve over time. This is done through a novel formulation and solution approach that leverages survival analysis estimates of conditional hazard rates, that also evolve over time (hazard rate processes). We show with new theory that these processes play a central role in determining optimal times to intervene. Indeed we show that optimal intervention is an optimal stopping problem on a hazard rate process. Significant theoretical insight into the structure of the optimal policy is also obtained.
>
> The work by Ren does not do any of this. The work of Ren et,. al. produces a single hazard rate function (so it is NOT a hazard rate process that evolves over time). Also, no new theoretical insight or result concerning hazard rate processes is developed.
>
> A secondary contribution of our work is an architecture (we call DDRSA) that adapts the architecture of Ren to model harzard rate processes from  co-variates that evolve over time using an encoder-decoder seq2seq approach.
>
>
> > what the relationship is between the covariates of the process and the event time L...
>
> At any time $j$ the relationship that $L$ has to the covariates ${\mathcal X}_j$ is encoded by the hazard rate function $h_j(k)$ which is actually dependent on the covariates as shown in equation (2). We have now included the training details and loss function to learn $h_j(k)$ using maximum likelihood (the supervised learning problem) into the main paper from the supplement.
>
>
> > Some of these details are in the appendix already, but it would be helpful to bolster the formalization and have them in the main paper.
>
>  As suggested, we have revised the paper to move most of Section B in the Appendix describing practical details of architecture, training and inference into the main paper. In addition we have added explanations connecting the math to the practical implementation. See section 4.1 and 5.1 in the revision.
>
> > It doesn’t seem that one can “ask” for a specific probability of failure
>
> Correct. Unfortunately, there is no direct way of setting $C_{\alpha}$ to guaruntee a particular $\alpha$. We are forced to try a range of values and pick the value with most acceptable tradeoff. We have added this to limitations.
>
> > I think that the experiments could be a lot more compelling if there were a plot showing the trade-off between the probability of failure and the expected time to intervene so that one could see visually what shape this trade-off curve takes (by setting different values of $C_{\alpha}$).
>
> As suggested, we have bolstered the paper with 10-fold CV tradeoff plots having $125$ increments of $C_{\alpha}$ between $8$ and $256$. These show that OTI method (which unlike static thresholds is not even tuned on a particular $C_{\alpha}$) outperfoms the static threshold algorithms over a broad range of probabilities of failure.

---

> > ### Comment · Reviewer_mWmk · 2022-08-08
> > **Thanks**
> >
> > Thanks for replying. The new figures look very informative. The other reviewers have brought up good points about clarity that I hope can be addressed. I have a remaining question.
> >
> > "At any time $j$ the relationship that $L$ has to the covariates $\mathcal{X}_j$ is encoded by the hazard rate function"
> >
> > So presumably this function is quite complex if it is allowed to depend on the entire history. I'm skeptical then about the start of Section 4 where it's claimed the RNN process is Markovian and captures the whole history. Is this meant to be an informal statement? I don't think this is true generally given the potential complexity of $h$.

---

> > > ### Author Response · Authors · 2022-08-09
> > > **Clarification on Markovian assumptions for RNN**
> > >
> > > >"At any time  the relationship that $L$  has to the covariates ${\mathcal X}_j$ is encoded by the hazard rate function"
> > >
> > > >So presumably this function is quite complex if it is allowed to depend on the entire history. I'm skeptical then about the start of Section 4 where it's claimed the RNN process is Markovian and captures the whole history. Is this meant to be an informal statement? I don't think this is true generally given the potential complexity of $h$
> > >
> > > This statement is not an informal statement and is made in the context of a feedforward RNN.
> > >
> > > The **hidden-state process**  of random variables $Z_j$ with time-index $j$ of the RNN is Markovian. From the feedforward RNN dynamics, the probability of the next hidden state is conditionally independent of the past hidden states, given the current hidden state. Specifically, we have:
> > >
> > > $P(Z_{j+1} | Z_j, Z_{j-1}, Z_{j-2},... ) = P(Z_{j+1} | Z_j), \forall j$
> > >
> > > In other words, changing the past hidden states will have no impact on the next RNN hidden-state, once we are given the current RNN hidden-state.
> > >
> > > Note, however, that we do not require the co-variates ${\mathcal X}_j$ to be Markovian. Indeed, as you correctly state, the hazard rate functions $h_j(k)$ can depend on the history of co-variates in general. Feedforward RNNs compute feature transformations of past covariate history (prior to $j$) and seek to embed all required information about the past into $Z_j$, so a hazard rate function that is computed from  $Z_j$ can indeed be quite complex and include (to the extent it is present in $Z_j$) information from past covariates ${\mathcal X}_j$.
> > >
> > > Note that this is not true of all RNN architectures (ex: Bidirectional RNNs). We have revised to clarify this: "In fact, the class of hazard rate processes that can be modeled by feedforward recurrent neural networks (RNNs), where the conditional hazard rate function at time step $j$ is computed only from the hidden-state $Z_j$, also directly satisfy our Markovian assumptions".

---

> ### Author Response · Authors · 2022-08-07
> **Rebuttal revision**
>
> Dear reviewer,
>
> We have updated the draft incorporating your suggestions.
>
> Specifically we:
> * Point out the dependence of the event time and the objective on the co-variates (Section 2, line 152-153, Section 2.1, lines 173-174)
> * Incorporated details of DDRSA architecture, loss, training and inference details from Appendix into main paper as suggested (Section 4.1)
> * Added discussion on the setting of $C_{\alpha}$ to the limitations (Section 5.2)
> * Generated tradeoff plots with fine sweep of $C_{\alpha}$  to show the shape of tradeoff curve as requested. (Section 5.2, Figs 3a and 3b)
>
> As the end of the discussion period is fast approaching, please do let us know if you feel other changes or clarifications are needed.

---

### Official Review · Reviewer_sQwG · 2022-07-16

**Rating:** 7
**Confidence:** 3
**Soundness:** 3 good
**Presentation:** 3 good
**Contribution:** 3 good

**Summary:**

This paper studies the task of finding optimal intervention time for failures. The problem can be viewed abstractly as a sequence of stochastically generated features. A failure happens at some point for this task, and the goal is to intervene optimally. This is done by a policy that either allows the process to run or decides to intervene at a given time. Optimal intervention is framed as deciding to intervene such that one does not intervene too early before the actual failure event, while also ensuring that the probability of late intervention (i.e., intervention after the failure) is below a threshold.

Two processes are then defined: a survival process and a hazard process. The survival process models the probability that failure will happen after a certain time from the current time. The hazard process $h_j(k)$ models the probability that failure will happen at a given time $j+k$ in the future when the current time is $j$. The optimal intervention time problem is then framed as finding the optimal stopping problem on the survival/hazard process. Under assumption that features evolve using Markovian dynamics, the optimal stopping problem can be solved.

Experiments are presented on two datasets and compare with two baselines. The model used for prediction is a  Recurrent Neural Network that converts a history of features $\{X_1, \cdots, X_j\}$ into a vector representation. This vector representation is used by a decoder to generate a sequence of vectors $\{Z_{jk}\}_{k}$ that is then directly converted to the hazard rates $h_j(k)$. Experiments show that broadly the new approach outperforms the baselines on both dataset.



**Questions:**

1. How is the DRSA loss computed? Is my understanding correct that the model is trained using DRS, then kept fixed, and used to solve equations in theorem 3 to find the optimal stopping time? As mentioned earlier, these experimental details would be helpful.

2. How will be $C_\alpha$ set in practice. This hyperparameter is less intuitive than $\alpha$ which might be easy to set based on domain knowledge.

3. Is there any particular reason why $(L-\tau)^+$ loss is used instead of square loss or any other loss?

4. Why is TTE seem to be doing better than OTI for large values of $C$ in Figure 3a?

Minor presentation:

1. "Corrolary" spelling error on Line 294. It should be "corollary"

2. Space in "Allmodels" on Line 320

3. In Figure 1, the sub-captions for the two figures seem to be wrong.

**Strengths And Weaknesses:**

**Strength:**
1. Problem is relevant to machine learning community from both theoretical and real-world impact
2. Solution is novel and well-motivated
3. Paper is mostly well-written

**Weakness**
1. The paper can use more details on the experiments. E.g., how the RNN model is used to make predictions will be useful and how is the RNN model trained, and are the equations Theorem 3.1 solved exactly or approximately? Details of computational aspects can also be useful since if the approach takes a long time to predict then it may be not useful for real-world applications where intervention must be done in real-time.

2. Discussion of Theorem 3 would be useful. Perhaps the other theorem can be removed to make space to discuss theorem 3 in more detail. E.g., a proof sketch can be provided, or a discussion of the theoretical results.

---

> ### Author Response · Authors · 2022-08-02
> **Response to reviewer**
>
> Thank you for noting the novelty, theoretical, and real-world impact of our work. We hope we were able to fully address your questions and mitigate perceived weaknesses.
>
> # Response to Questions
>
> > How is the DRSA loss computed?
>
> This was given in Appendix B (pg 6 and 7 of the supplement). We have now revised the paper to move the detailed discussion of DDRSA architecture, training details including computation of loss function into the main paper itself (Section 4.1).
>
> > Is my understanding correct that the model is trained using DRSA, then kept fixed, and used to solve equations in theorem 3 to find the optimal stopping time? As mentioned earlier, these experimental details would be helpful.
>
> You are correct! Once the DDRSA model is trained, we estimate, at any time-index $j$,  $h_j(0), h_j(0), \cdots h_j(L_{max}-1)$ by passing values of ${\mathcal X}_j$ into the DDRSA architecture (See Figure 1 on page 6 of the supplement) or Section 4.1 of the revision. Then, we calculate expected time to event using equation (12) and the value function using equation (11). Intervention is triggered at time $j$ if expected time to event falls below the value function. We have added this description of the inference process to Section 4.1 of the revision.
>
> > As mentioned earlier, these experimental details would be helpful (inference times etc.).
>
> Details of computational setup are in Appendix D of the supplement. We have also added inference time of DDRSA and policy computation times into the main revised paper (Section 5.2).
>
> > How will $C_{\alpha}$ be  set in practice. This hyperparameter is less intuitive than  which might be easy to set based on domain knowledge.
>
> Unfortunately, there is no direct way of setting $C_{\alpha}$ to guarantee a particular $\alpha$. We sweep through a range of values of $C_{\alpha}$ and pick the value with most acceptable trade off. We have added this to the limitations section.
>
> > Is there any particular reason why $(L-\tau)^{+}$ loss is used instead of square loss or any other loss?
>
> $E[ (L - \tau)^{+} ]$ is mean residual life, a well known and important quantity in survival analysis [See chapter 4 of Chin-Diew Lai, Min Xie, Stochastic Aging and Dependence for Reliability] (https://link.springer.com/chapter/10.1007/0-387-34232-X_4?noAccess=true). It can be expressed using hazard rates, besides being highly relevant to the problem as the expected residual time to the critical event.
>
>
> > Why is TTE seem to be doing better than OTI for large values of $C_{\alpha}$ in Figure 3a?
>
> There are a few possible reasons for this:
>
> * Noise with 5-fold estimation with a small sample test set.
>
> * Our approach requires fitting a survival model for $h_j(k)$ at each time step over several values of $k$. It is possible that some of these are better estimated than others, while in contrast a threshold algorithm can be tuned for a specific $C_{\alpha}$ on a validation set. What we expect with OTI, however is a broad range of miss-rates at which it will outperform. Being able to weight the survival estimation for enhanced policy performance is an interesting next step. We have added this to limitations in Section 5.2.
>
> * At high values of $C_{\alpha}$ the competing models tuned on small validation and test sets will tend to underestimate the risk of seeing a critical event. Consider a case where a certain intervention threshold acheives $0\%$ miss rate on the validation set. Then as we increase $C_{\alpha}$ the same intervention threshold will continue to be selected. The correct behavior, of course, is to predict earlier by lowering the intervention threshold. If the test set is small and missing "tail events" to truly test the model, the performance will appear better than in truth.
>
> Note that in the revision we have included 10-fold CV results with broad $C_{\alpha}$ sweep as requested by one of the other reviewers. The OTI policy outperforms over a broad range of miss-rates, even though it is not explicitly tuned for a $C_{\alpha}$, just by plugging estimated hazard rates into the optimal intervention policy formula.
>
> >A discussion of Theorem 3 (proof sketch)
>
> Included in the revision.

---

> > ### Comment · Reviewer_sQwG · 2022-08-08
> > **Thank you**
> >
> > Thank you for the detailed response. I have increased the score.

---

> ### Author Response · Authors · 2022-08-07
> **Rebuttal revision**
>
> Dear reviewer,
>
> We have updated the draft incorporating your suggestions.
>
> Specifically we:
> * Added proof sketches (Proposition 2.1 and Theorem 3.1)
> * Incorporated details of DDRSA architecture, loss, training and inference details from Appendix into main paper (Section 4.1)
> * Provided details of computational setup and inference times (Section 5.2)
> * Added discussion on the setting of $C_{\alpha}$ to the limitations (Section 5.2)
>
> As the end of the discussion period is fast approaching, please do let us know if you feel other changes or clarifications are needed.

---

### Author Response · Authors · 2022-08-02
**Summary of revisions**

* Removed parallel discussion of survival processes, retaining the hazard rate process point of view only (reviewer KFJC)
* Section 1.1 and I.2: Included discussion of special cases, added clarifying language (reviewer KFJC)
* Proposition 2.1: Added proof sketch (reviewer sQwG) and pointed out dependence on co-variates (reviewer KFJC)
* Theorem 3.1: Separated stopping time from policy, removed mention of homogeneity and stationary policies (reviewer KFJC) and added proof sketch (reviewer sQwG)
* Section 4.1 Incorporated details of DDRSA architecture, loss, training and inference details from Appendix into main paper (reviewer SQWG, mWmk)
* Section 5.2: Regenerated tradeoff plots with finer sweep of $C_{\alpha}$ to show shape of tradeoff curve (reviewer mWmK, KFJC)
* Section 5.2:Computational setup and inference times provided
* Section 5.2: Expanded discussion of limitations including the setting of $C_{\alpha}$ (reviewer sQwG, mWmK, KFJC, vH1X)

---

### Meta-Review · Area_Chair_MJxr · 2022-08-26

**Recommendation:** Accept
**Confidence:** Less certain

**Metareview:**

The reviewers all appreciate the direction of this work, and while the merits and significance of the work have limitations -- as reflected in the weak scores -- all reviewers were positive and found no reason to reject. I agree with this and recommend acceptance on the basis that the merits of the contribution and having this as part of the program outweigh the concerns regarding significance. That said, I strongly encourage the authors to use the detailed feedback from the reviewers to improve their paper.

**Award:**

No

---

### Decision · Program_Chairs · 2022-09-14

Accept